# Incomplete to complete multiphysics forecasting - a hybrid approach for learning unknown phenomena

## Abstract

Modeling complex dynamical systems where only partial knowledge of their physical mechanisms is available is a crucial problem across all scientific and engineering disciplines. Purely data-driven approaches, which only make use of an artificial neural network and data, often fail to accurately simulate the evolution of the system dynamics over a sufficiently long time and in a physically consistent manner. Therefore, we propose a hybrid approach that uses a neural network model in combination with an incomplete PDE solver that provides known, but incomplete physical information. In this study, we demonstrate that the results obtained from the incomplete PDEs can be efficiently corrected at every time step by the proposed hybrid neural network – PDE solver model, so that the effect of the unknown physics present in the system is correctly accounted for. For validation purposes, the obtained simulations of the hybrid model are successfully compared against results coming from the complete set of PDEs describing the full physics of the considered system. We demonstrate the validity of the proposed approach on a reactive flow, an archetypal multi-physics system that combines fluid mechanics and chemistry, the latter being the physics considered unknown. Experiments are made on planar and Bunsen-type flames at various operating conditions. The hybrid neural network - PDE approach correctly models the flame evolution of the cases under study for significantly long time windows, yields improved generalization, and allows for larger simulation time steps.

## 1 Introduction

Modeling and forecasting of complex physical systems described by nonlinear partial differential equations (PDEs) are central to various domains with applications ranging from weather forecasting (Kalnay, 2003), design of airplane wings (Rhie & Chow, 1983), to material science (Wheeler et al., 1992). Typically, a chosen set of PDEs are solved iteratively until convergence of the solution. Modeling complex physical dynamics requires a good understanding of the underlying physical phenomena. For cases where the complete physics information is missing, deep learning models can be employed to complete the physical description when additional data of the system is available. Deep learning methods have shown promises to account for these unknown components of the system (Yin et al., 2021). We consider a set of partial differential equations with partially unknown physics represented. The corresponding PDE model for a general state $\phi$ is given by

$$\frac{\partial \phi}{\partial t} = \mathcal{P}_i(\phi, \frac{\partial \phi}{\partial x}, \frac{\partial^2 \phi}{\partial x^2}, ...) + S_\phi, \tag{1}$$

where $\mathcal{P}_i$ models the known but incomplete PDE description, $S_\phi$ represents the unknown physics.

Within the scope of this model, the influence of $S_\phi$ can lead to fundamentally different scenarios. A commonly targeted case is when the governing equations of the complete PDE description are computationally too expensive to solve, turbulence modeling in computational fluid dynamics (CFD) being a good example. In CFD, a spacial filtering is performed on the original governing PDEs. This step introduces unclosed terms in the model equations that correspond to unrepresented physics in equation 1, due to the effects of the filtered scales. This is a widely studied problem, where the use of deep learning models is currently being explored (Lapeyre et al., 2019; Kochkov et al., 2021; List

et al., 2022; Stachenfeld et al., 2022). In the following, we are targeting a more challenging problem, where increasing spacial resolution and/or reducing time-scales of the incomplete PDE solver does not lead to a converged full solution. Rather, the incomplete and complete PDEs produce drastically different solutions due to the unknown physics. The central learning objective is to correct this behavior and retrieve the evolution that would be obtained with the complete PDE description.

Our work expands on the combination of incomplete PDE solvers and neural networks (NNs) (Yin et al., 2021; Takeishi & Kalousis, 2021) to account for the effects of an incomplete physics model. The NN aims to complete the PDE description, where the differences in complete and incomplete PDE solutions are beyond the effects of spacial and temporal scales. We showcase that combining the trained NN model with a differentiable solver for the incomplete PDE can accurately reproduce the physical solutions of the complete, multi-physics PDE solver with stable long-term rollouts.

We demonstrate the capabilities of this approach for an archetypal multi-physics system, namely a complex reactive flow which combines fluid mechanics and chemistry, where the latter is considered unknown. Reactive flow modeling has applications in numerous domains such as combustion processes in gas turbines (Lieuwen, 2012), climate modeling (Jacobson, 1999; Rolnick et al., 2022) and astrophysics simulations (Gamezo et al., 2003). Resolving the Navier-Stokes equations lies at the core of these problems, where additionally the transport of different species of relevance must also be accounted for, together with their production or consumption often following complex reaction mechanisms (Poinsot & Veynante, 2005). For chemically reacting flows, generation or consumption of multiple species via some chemical reaction are modeled using a net source term. It is a well-known fact that the incorporation of a detailed chemical kinetic mechanism in a reacting flow model can result in a stiff system of governing equations (Wanner & Hairer, 1996; Najm et al., 1998; Knio et al., 1999).

We consider the reactive flow simulation to be the complete PDE description, while the non-reactive flow simulation represents the incomplete PDE basis, where the chemical kinetic mechanisms are collected in the unknown physics term of equation 1. Figure 1 shows a visual example of the fundamental differences in system dynamics that can be caused by unknown reaction terms. We showcase the effectiveness of our approach for different cases of planar 2D premixed methane-air flames, and the varying transient evolution of Bunsen-type flames. We show that the proposed approach can handle large domains with highly resolved flames, which is closer to the practical flame domains used in many industrial applications. Specifically we concentrate on training a NN model to correct the spatio-temporal effects of energy and species transport source terms. We show that in addition to recovering the desired solutions,

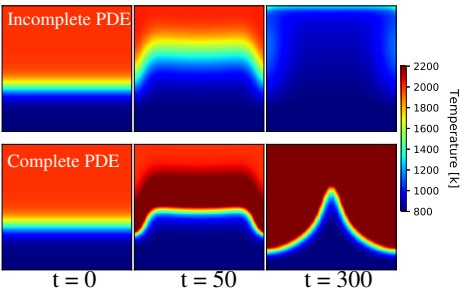

Figure 1: The incomplete/non-reactive (top) and complete/reactive (bottom) PDE solvers we consider can yield fundamentally different evolutions, as shown here for a sample temperature field over time.

this approach overcomes inherent problems of temporal stiffness due to the complex reaction mechanism. Lastly, we demonstrate the applicability of neural network model to control the flow inlet velocities to arrive at desired flame shapes, when combined with a differentiable flow solver.

## 2 RELATED WORK

Deep learning methods have been widely used to model the solutions of partial differential equations (Lagaris et al., 1998; Long et al., 2018; Han et al., 2018; Bar-Sinai et al., 2019) and in particular, the Navier-Stokes equations (Thuerey et al., 2020; Fukami et al., 2019). These models can be very fast and do not suffer from the time-step stability issues associated with traditional numerical solvers. Nevertheless, as these purely data-driven approaches lack the physical understanding of the system being modeled, they generally fail in generalizing to other operating conditions (Kim et al., 2019; Lapeyre et al., 2019). To leverage the potential of deep learning in physical simulations, it is therefore necessary to incorporate some physical information within the deep learning framework.

Deep learning models can enforce physical constraints partially through the loss function (Bar-Sinai et al., 2019; Raissi et al., 2019) or changes in neural network architecture (Greydanus et al., 2019; Lu et al., 2021). However, these approaches struggle to enforce physical constraints such as boundary conditions or predict the strong unsteadiness and chaotic nature of flows. Neural operators (Li et al., 2020b; Bhattacharya et al., 2021; Patel et al., 2021), specifically, the Fourier neural operator (FNO) of Li et al. (2020a) introduce an interesting line of work by learning mesh-free, infinite-dimensional operators with neural networks, but do not necessarily offer advantages for longer term predictions.

Hybrid approaches that combine machine learning techniques with numerical PDE solvers (Wang et al., 2020), have attracted a significant amount of interest due to their capabilities for generalization (Chen et al., 2018). In this context, neural networks are typically used to model or replace a part of the conventional PDE solver to improve aspects of the solving process. For example, Tompson et al. (2017), Özbay et al. (2021) and Ajuria Illarramendi et al. (2020) proposed a convolutional neural network based approach to solve the Poisson equation in CFD simulation. In recent years, a number of deep learning based models have been introduced to accurately model turbulent flows (Pathak et al., 2020; Stachenfeld et al., 2022; Dresdner et al., 2022). Um et al. (2020) and Belbute-Peres et al. (2020) showed the advantages of training neural networks with differentiable physics to correct the numerical errors that arise in the discretization of PDEs. Sirignano et al. (2020) and Kochkov et al. (2021) additionally correct for closure error by integrating neural network with a differentiable CFD simulator. These approaches demonstrate the capabilities of neural networks to correct errors in a fast, under-resolved simulation.

Similar to the goals of our work, Yin et al. (2021) and Takeishi & Kalousis (2021) have introduced frameworks of augmenting incomplete physical dynamics with neural network models. These approaches demonstrate the applicability on ODE/PDE systems, which are weakly nonlinear and the unknown dynamics are linear combinations of the underlying flow fields. We expand on these works to explore the significantly more challenging scenario of reactive flows. These are characterised by a multi-physics system with nonlinear advective terms and strongly nonlinear dynamics, described by exponential source terms that exhibit nonlinear combinations of the flow fields.

## 3 METHODOLOGY

We consider two different sets of PDEs with their associated numerical solver, which we denote as the *incomplete* PDE $\mathcal{P}_i$ and the *complete* PDE $\mathcal{P}_c$. By evaluating $\mathcal{P}_c$ on an input state $\phi_t$ at time $t$, we can compute the points of the phase space sequences; $\phi_{t+\Delta t} = \mathcal{P}_c(\phi_t)$. Without loss of generality, we assume a fixed time-step $\Delta t$ and denote a state $\phi_{t+\Delta t}$ at next time instance as $\phi_{t+1}$. The main goal of this study is to learn a correction function that models the effects of the unknown physical system on the incomplete PDE solver to obtain the complete PDE solutions. We train a correction model $C(\phi; \theta) : \mathcal{X} \to \mathcal{X}$ that maps $\mathcal{P}_i(\phi_t)$ to $\phi_{t+1}$, defined over a finite time interval $[0, T]$ and a spacial domain $\Omega \subset \mathbb{R}^2$. The learning objective is to find the best possible correction function $C(\phi; \theta)$ given the weights $\theta$ and an input flow state, $\phi$. The model parameters $\theta$ are estimated from the complete PDE solution trajectories $(\phi_0, \phi_1, .., \phi_T)$. The learned predictions obtained after repeatedly applying the corrector $C$ and invoking $\mathcal{P}_i$ are denoted by $(\tilde{\phi}_0, \tilde{\phi}_1, .., \tilde{\phi}_T)$.

### 3.1 PARTIAL DIFFERENTIAL EQUATIONS FOR REACTIVE FLOWS

The physical system we investigate is a laminar premixed methane-air flame with a single step chemistry. It is governed by the following Navier-Stokes equations (Poinsot & Veynante, 2005)

$$
\begin{aligned}
\frac{\partial \rho}{\partial t} + \frac{\partial \rho u_i}{\partial x_i} &= 0 & \rho C_p \left( \frac{\partial T}{\partial t} + \frac{\partial u_i T}{\partial x_i} \right) &= \dot{\omega}'_T + \frac{\partial}{\partial x_i} \left( \lambda \frac{\partial T}{\partial x_i} \right) \\
\frac{\partial \rho u_j}{\partial t} + \frac{\partial \rho u_i u_j}{\partial x_i} &= -\frac{\partial p}{\partial x_j} + \frac{\partial \tau_{ij}}{\partial x_i} & \frac{\partial \rho Y_k}{\partial t} + \frac{\partial \rho u_i Y_k}{\partial x_i} &= \dot{\omega}_k + \frac{\partial}{\partial x_i} \left( \rho D_k \frac{\partial Y_k}{\partial x_i} \right)
\end{aligned}
\tag{2}
$$

where $\rho, u_i, p, T, Y_k$ denote the density, velocity, pressure, temperature, and species mass fractions of species $k$, respectively. $\tau, D_k, \lambda$ are the strain rate tensor, the diffusion coefficient of species $k$ and mixture thermal conductivity. In addition, $C_p$ denotes the mixture specific heat capacity. $\dot{\omega}_k$ and $\dot{\omega}'_T$ are the species reaction rate and heat release rate, respectively.

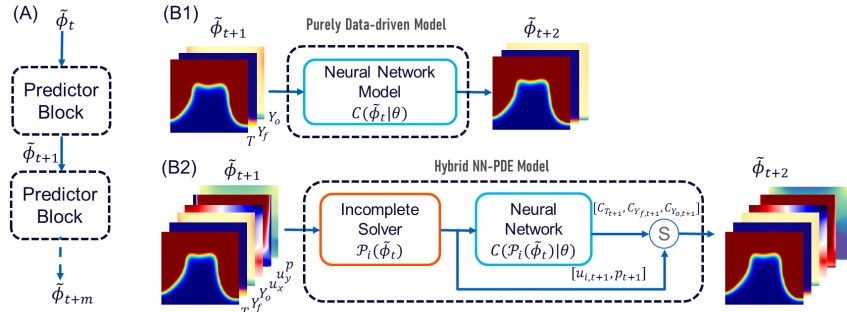

Figure 2: (A) Multi-step training framework helps to learn the dynamics of complete PDE solver over longer rollouts. (B1) Details of the input flow state and predictor block used in a purely data-driven approach and (B2) the hybrid NN-PDE approach, where S denotes the concatenation of different fields to obtain the complete flow state $\tilde{\phi}$ at next time step.

We consider a one-step chemical system of the type $\upsilon'_F F + \upsilon'_O O \rightarrow$ products, where $\upsilon'_F$ and $\upsilon'_O$ are the stoichiometric coefficients corresponding to fuel and oxidizer. The simplifications proposed by Williams (1985) and Mitani (1980) are used to model the reaction rates $\dot{\omega}_k$ for each species. The fuel consumption rate $\dot{\omega}_F$ is assumed to have the Arrhenius form.

$$\dot{\omega}_F = \upsilon'_F W_F B_1 T^{\beta_1} \left(\frac{\rho Y_F}{W_F}\right)^{n_F} \left(\frac{\rho Y_O}{W_O}\right)^{n_O} \exp\left(-\frac{E_a}{RT}\right). \tag{3}$$

The heat release source term $\dot{\omega}'_T$ and the fuel source term $\dot{\omega}_F$ are linked by $\dot{\omega}'_T = -Q\dot{\omega}_F$, where $Q$ is the heat of reaction per unit mass. Following Poinsot & Veynante (2005), parameters corresponding to a real-world methane-air flame are chosen as: $B_1 = 1.0810^7$ uSI; $\beta_1 = 0$; $E_a = 83600$ J/mole; $n_F = 1$; $n_O = 0.5$; $Q = 50100$ kJ/kg; $C_p = 1450$ J/(kgK). Taken together, the system of equations above is a challenging scenario even for classical solvers, and due to its practical relevance likewise a highly interesting environment for deep learning methods.

## 3.2 PROBLEM FORMULATION

The incomplete PDE solver solves the set of equations 2 without the source terms and reaction rates $\dot{\omega}_k$ and $\dot{\omega}'_T$, while the complete PDE solver solves the full set from equation 2. The neural network model, denoted by $C(\mathcal{P}_i(\phi)|\theta)$, corrects the incomplete/non-reacting flow states $\mathcal{P}_i(\phi)$ to obtain the complete/reacting states $\mathcal{P}_c(\phi)$ as shown in figure 2 (B2). The neural network is trained to model the effects of the unknown chemistry using parameters $\theta$ given an input flow state, $\phi = [u_i, p, T, Y_f, Y_o]$. As seen from equation 2, and equation 3, the temperature and species mass fractions fields are strongly coupled which significantly increases the prediction problem complexity. A slight error in one of the fields will quickly propagate into the other fields, making the predictions diverge. In the following, a subscript $C_s(\circ)$ will denote that the neural network $C$ generates the field $s$, e.g., $C_T$ generating the temperature field. $\tilde{\phi}_{t+1} = [u_{i,t+1}, p_{t+1}, C_T(\mathcal{P}_i(\tilde{\phi}_t)|\theta), C_{Y_f}(\mathcal{P}_i(\tilde{\phi}_t)|\theta), C_{Y_o}(\mathcal{P}_i(\tilde{\phi}_t)|\theta)]$ where $\tilde{\ }$ indicates a corrected state. $u_{i,t+1}$ and $p_{t+1}$ are the time-advanced velocity and pressure field, respectively, predicted using the incomplete PDE solver.

## 3.3 TRAINING METHODOLOGY

We employ a *hybrid NN-PDE* approach that augments a neural network model with a PDE solver (Um et al., 2020; Kochkov et al., 2021). In contrast to previous work, we use the incomplete PDE solver as a basis, and hence the solver does not converge to the desired solutions under refinement, as explained in 3.1. The neural network is integrated and trained in a loop with the incomplete PDE solver using stochastic gradient descent for $m$ time steps, as shown in figure 2 (A,B2). Here, the number of temporal look-ahead steps, $m$, is an important hyper-parameter of the training process. Higher $m$ provides the network with longer-term feedback at training time through the gradient rollouts. This gives the model improved feedback on how the time dynamics of the incomplete PDE solver affect the input states, and hence which corrections need to be inferred by the model.

A central component of the hybrid NN-PDE model is the differentiable solver, which allows us to embed the solver for the incomplete PDE system in the training of a neural network. The differentiable solver acts as additional non trainable layers in the network. Instead, they provide derivatives of the outputs of the simulation with respect to its inputs and parameters. Here, we use the differentiable PDE solver from the *phiflow* framework of Holl et al. (2019) in combination with *Tensorflow* to obtain the non-reactive flow solver and reactive flow solver solutions. The marker-and-cell method (Harlow & Welch, 1965) is adopted to represent temperature, pressure, density, species mass fraction fields in a centered grid, and velocities in a staggered grid.

A *purely data-driven* model (PDD) is used as a baseline. It employs a neural network model to learn the complete flow states $\mathcal{P}_c(\phi)$ given an input flow state $\phi^S$ where $\phi^S = [T, Y_f, Y_o]$, as shown in figure 2 (B1). The new state predicted by the trained neural network model is $\tilde{\phi}_{t+1}^S = C(\tilde{\phi}_t^S | \theta)$. For the hybrid NN-PDE as well as PDD, the neural network part of the predictor block in figure 2 consists of a fully convolutional neural network model. For both, we experimented with both the ResNet (He et al., 2016) and UNet (Ronneberger et al., 2015) architectures. We found that the ResNet performed the best for the PDD setting, while for the hybrid NN-PDE approach the UNet performed consistently better. Hence, the following results will use a ResNet for PDD models, and a UNet for the NN-PDE hybrids. Additional details of the neural network architectures are provided in the appendix. The network receives the input states as shown in figure 2. The output of the neural network model is used to obtain the corrected state $\tilde{\phi}_{t+1}$ as specified above. We constrain the mass fraction fields $Y_k$ to contain physical values in the range $Y_f \in [0, 0.05]$ and $Y_o \in [0, 1]$. All models are trained for 100 epochs with a batch size of 3 and a learning rate of 0.0001.

We additionally compare against the *Fourier neural operator* (FNO) of Li et al. (2020a) as an example of a state-of-the-art neural operator method. The new state predicted by the trained model is given by $\tilde{\phi}_{t+1}^S = C(\tilde{\phi}_t^S | \theta)$ where $\tilde{\phi}^S = [T, Y_f, Y_o]$ and $C(\circ)$ represents the Fourier neural operator. All three approaches use an $L_2$ based loss that is evaluated for $m$ steps as $\mathcal{L}(\theta) = \sum_{n=t+1}^{t+m} \sum_{\phi=\{T, Y_f, Y_o\}} ||\tilde{\phi}_n, \mathcal{P}_c(\phi_n)||_2$ .

## 4 NUMERICAL EXPERIMENTS

We consider a case of planar 2D premixed methane-air flame propagating in a quiescent mixture (**Planar-v0**) and two cases of the transient evolution of initially planar laminar premixed flame into a Bunsen-type flame under different inlet velocity conditions (**uniform-Bunsen, nonUniform-Bunsen**). We obtain the target data by considering the source terms as defined by equation 3 with the parameters mentioned in section 3.1.

**Planar-v0**    For the most basic case, the planar 2D flame model setup, we consider the reacting Navier-Stokes equations described in section 3 with zero inlet velocity i.e. $u_x = 0$, $u_y = 0$. We consider a square domain of size 0.05 m × 0.05 m with 32 × 32 resolution and closed boundary conditions. The simulation is initialized using a steep transition between a premixed methane-air mixture and burnt gases. Our training data consists of 6 simulations of 300 steps created by varying the equivalence ratio $E$. It represents the stoichiometric mixture ($\varphi$) of fuel $Y_f$ and oxidizer $Y_o$ mass fractions, i.e., $E = \varphi \frac{Y_f}{Y_o}$ and thus fundamentally influences the dynamics of the chemical reaction. For the training data we use $E_{train} = \{1.0, 0.9, 0.8, 0.7, 0.6\}$, while the test dataset contains $E_{test} = \{0.95, 0.85, 0.75, 0.65\}$.

**uniform-Bunsen**    In contrast to the planar case, the premixed methane-air mixture is now fed with a constant inlet velocity. The boundary conditions upstream, at $y = 0$, are $(u_x, u_y)_{x, y=0} = (0, \kappa)$ where $\kappa$ is a $n$-dimensional vector with constant amplitude. The target simulation contains a heat release rate term, and in this case the initial temperature field evolves into different $\Lambda$-shaped flames at the end of the $300^{th}$ time step. Training and testing datasets are created by varying the equivalence ratio and inlet velocity amplitude $U$: $E_{train} = \{1.0, 0.9, 0.8\}$ and $U_{train} = \{0.45, 0.4, 0.3\}$. The test dataset uses $E_{test} = \{0.95, 0.85\}$ $U_{test} = \{0.43, 0.375, 0.325\}$. The length and temperature of the flame significantly vary depending on the inlet velocity and equivalence ratio provided. Further details on the boundary conditions and training data visualizations can be found in the appendix.

**nonUniform-Bunsen**    As a third case we consider the transient evolution of a premixed methane-air flame with non-uniform inlet velocity. The boundary conditions upstream, at y = 0, are

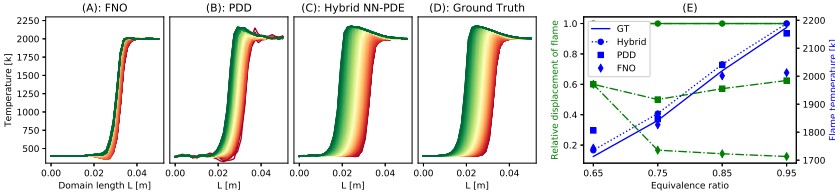

Figure 3: 1D cut of the planar flame simulation over 300 steps. The initial state is plotted in red, target state in green. Hybrid NN-PDE approach predicts physically accurate results over longer rollouts with correct flame temperature and relative displacement of flame front across different equivalence ratios.

Table 1: Mean and standard deviation of errors over all time steps of all testsets. The Hybrid NN-PDE approach outperforms all other baselines considered.

|  |  | Baseline FNO | Baseline NN | Hybrid NN-PDE | Hybrid NN-PDE-dt |
|---|---|---|---|---|---|
| MAPE | Planar-v0 | $8.27 \pm 5.51\%$ | $6.33 \pm 3.05\%$ | $1.40 \pm 0.65\%$ | **1.21** $\pm 0.39\%$ |
|  | uniform-Bunsen | $15.57 \pm 8.72\%$ | $7.58 \pm 3.73\%$ | **0.72** $\pm 0.37\%$ | $1.11 \pm 0.47\%$ |
|  | nonUniform-Bunsen32 | $12.30 \pm 7.98\%$ | $12.48 \pm 11.31\%$ | **2.04** $\pm 1.39\%$ | $2.46 \pm 1.61\%$ |
|  | nonUniform-Bunsen100 | - | - | **3.23** $\pm 3.76\%$ | $4.14 \pm 4.99\%$ |
| MSE | Planar-v0 | $88316 \pm 124394$ | $34491 \pm 43966$ | $1122 \pm 1300$ | **292** $\pm 296$ |
|  | uniform-Bunsen | $220817 \pm 167024$ | $72563 \pm 56919$ | **721** $\pm 1007$ | $1267 \pm 1622$ |
|  | nonUniform-Bunsen32 | $184860 \pm 185694$ | $130219 \pm 142094$ | **9647** $\pm 13903$ | $17569 \pm 24392$ |
|  | nonUniform-Bunsen100 | - | - | **23293** $\pm 37451$ | $26862 \pm 47639$ |

$(u_x, u_y)_{x,y=0} = (0, \kappa)$ where $\kappa$ is a $n$-dimensional vector whose elements are each sampled from a uniform distribution from [0.2, 0.65]. We experiment with two different domain sizes, $32 \times 32$ (**nonUniform-Bunsen32**) and $100 \times 100$ (**nonUniform-Bunsen100**), as shown in figure 4. The larger domain size used is closer to the practical reactive flow domain used in CFD applications (Jaensch et al., 2017) with a highly resolved flame. These inlet velocity conditions generate complex flame shapes which increase the difficulty of the prediction problem. We consider simulation sequences with 500 steps as it takes longer time for the flame to reach the steady-state solution.

## 5 RESULTS

We demonstrate the capabilities of the proposed learning approach to represent the complete PDE description with the aforementioned cases of increasing difficulty. We also study its ability to generalize to unseen operating conditions such as equivalence ratios, simultaneous variations in constant inlet velocity and equivalence ratio, and non-uniform inlet velocity profiles. The source code of our work will be published upon acceptance. As baselines, we compare against a purely data-driven approach; a neural network model with exactly the same look-ahead steps, and the Fourier neural operator of Li et al. (2020a) that likewise includes $m$ look-ahead steps as discussed in section 3.3.

### 5.1 QUALITATIVE AND QUANTITATIVE ANALYSIS

Table 1 compares the mean absolute percentage errors (MAPE) and mean squared errors (MSE) of temperature field for all the cases discussed in section 4. For Planar-v0, the FNO and PDD approaches yield large errors with a MAPE of 8.27% and 6.33%, respectively. On the other hand, the hybrid NN-PDE model trained with 32 look-ahead steps reduces the error to 1.4%, and thus performs significantly better than the two baselines. This behavior is visualized in figure 3 with a 1D transverse cut of the simulation domain over 300 time-steps. The hybrid NN-PDE approach successfully captures the propagation of the flame.

In figure 3 (E), we also compare two important physical quantities, the flame temperature and relative displacement of the flame front, across different equivalence ratios. It can be seen that the hybrid NN-PDE model (blue circles) accurately predicts the flame temperature for different test cases (solid

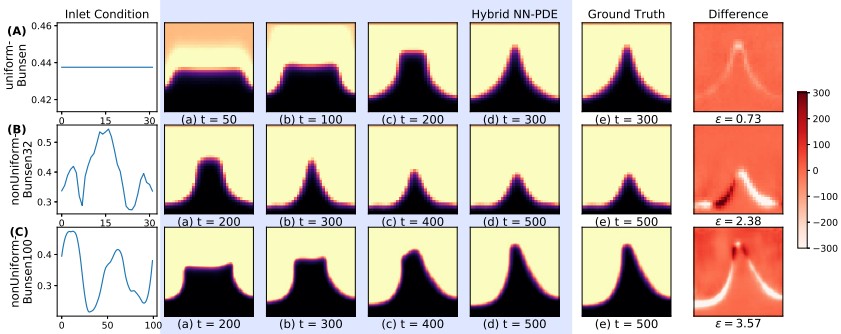

Figure 4: Left : Inlet velocity profile used. (a-d) Temperature field prediction by hybrid NN-PDE approach over different steps given the inlet velocity profile. (e) Ground truth data. Right : Difference between ground truth data and hybrid NN-PDE output at last snapshot. Top to bottom: $32 \times 32$ resolution cases of (A) uniform-Bunsen, (B) nonUniform-Bunsen32, and (C) $100 \times 100$ testcase nonUniform-Bunsen100. $\epsilon$ represents the instantaneous MAPE.

blue line). The relative displacement of the flame front is computed as $|\tilde{x}_t - \tilde{x}_0|/|x_t - x_0|$, where $x_t$ is the position along the flame normal at time $t$ on the $1200K$ isotherm of the ground truth simulation, and $\tilde{x}$ denotes the predicted position. For all test cases, the hybrid approach accurately predicts the flame displacement over $t = 300$ steps, while the other approaches yield significant errors.

For uniform-Bunsen case, amongst the baseline models, PDD performs better with an error of 7.58% compared to FNO model which yields an error of 15.57%. A neural network model combined with an incomplete PDE solver for 32 look-ahead steps yields a significantly lower error of 0.72% as shown in table 1. This means, the hybrid approach reduces the errors by a factor of 10 over the baselines considered. Figure 4(A) shows the temporal evolution of the hybrid NN-PDE approach predicted over 300 time steps for the uniform-Bunsen case. It shows the results for a test case with $U = 0.4375$ and $E = 0.95$. It predicts a symmetric flame with accurate flame height and achieves very low instantaneous MAPE of 0.73% at $t$=300, when compared with the ground truth data.

Next, we study a complex scenario of nonUniform-Bunsen flame with $32 \times 32$ resolution. The hybrid NN-PDE approach outperforms the PDD approach and FNO with an improvement of $\sim 80\%$. The MSE values show this trend even more clearly. The large standard deviations of the MSE numbers indicate that the predictions made by the baseline approaches contain substantial deviations from target values. Important to highlight is that, despite using the same training data and similar neural network architecture, with same look-ahead steps, the hybrid approach outperforms the PDD approach thanks to its learned collaboration with the incomplete PDE solver. It accurately reproduces the complete PDE behavior. Figure 4(B) highlights this with a visualization of the hybrid NN-PDE model predictions for nonUniform-Bunsen32 case. The trained model accurately predicts the flame simulation over long roll-outs of 500 steps and achieves the complex flame shape with a low, instantaneous MAPE of 2.38%. Finally, we showcase the ability of hybrid approach to predict the temporal evolution of highly resolved flames with the uniform-Bunsen100 scenario. Despite the increased complexity of the larger resolution, it achieves a very good overall MAPE of 3.23% over 12 test cases of 500 simulation steps. Figure 4(C) shows an example of physically accurate predictions made by the hybrid NN-PDE model. We omit the evaluation of baselines for high resolution cases as they do not succeed to model the flame dynamics for low resolution cases.

**Effect of training dataset size** Figure 5 (A) shows that increasing the number of training sets has little or no effect on the prediction capabilities of the PDD models for nonUniform-Bunsen32 case. The hybrid model with 12 training sets achieves a MAPE of $12.49 \pm 4.17\%$, an improvement of 38% over the PDD model with 32 training sets. This result strengthens the hypothesis that integrating the incomplete PDE solver into the neural network training yields a learning signal that fundamentally differs from that produced by training with precomputed data. The purely data-driven models cannot achieve the same level of accuracy even in the presence of large amounts of data.

**Generalization to incorrect PDE parameters** The proposed hybrid NN-PDE model is capable of completing the PDE description even if the underlying incomplete PDE solver does not have access

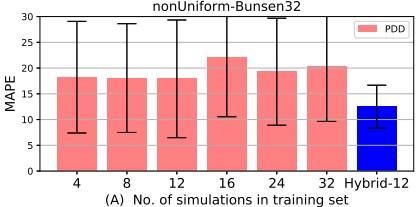
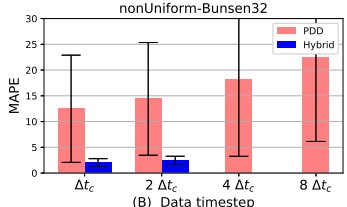

Figure 5: (A) Effect of increasing training dataset size for PDD approach, over fixed testsets, is compared with equivalent (trained with same look-ahead steps $m = 2$) hybrid NN-PDE model. (B) Effect of temporal coarsening on PDD models trained with $m = 32$ look-ahead steps.

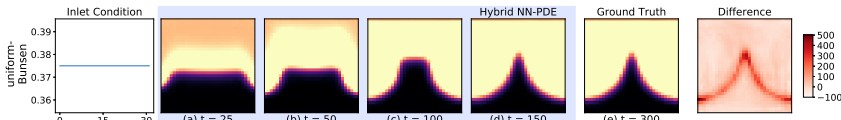

Figure 6: Left : Constant inlet velocity profile used. (a-d) uniform-Bunsen case trained and inferred at double time-step $2 \, \Delta t_c$, using hybrid NN-PDE approach. It achieves same flame shape at $t = 150$ for which the complete PDE solver requires 300 time-steps.

to the correct parameters, referred to as 'incomplete, incorrect PDE'. For this test we assume that the values of $\tau$, $D_k$, $\lambda$ in equation 2 are unknown and set to incorrect values. For the nonUniform-Bunsen32 case, the hybrid model with the incomplete, incorrect solver achieves an overall MAPE of $2.48 \pm 1.20$ %, a slight increase over the MAPE of $2.04 \pm 1.39$ % for the hybrid model with the incomplete, correct PDE. Additional details can be seen in appendix C.

## 5.2 RELAXING TEMPORAL STIFFNESS IN THE PDE SOLVER

Traditionally, the source terms and reaction rate terms involved in modeling the fast chemistry of reacting flows require the use of very small time-steps in simulations due to the stiffness of the chemical mechanisms. The incomplete PDE solver used in the hybrid NN-PDE approach, does not contain the source terms and reaction rate terms. Therefore, the time scales associated with the chemical reactions play a less important role in maintaining numerical stability, and it becomes possible for the solver to employ larger time steps. To illustrate this advantage, we train the neural network model with a time-step that is twice as large as the largest time-step $\Delta t_c$ required for the complete PDE solver to yield a stable result.

We mimic the setup described in section 4, but now the incomplete PDE solver uses a time-step of $2\Delta t_c$, which is too large for the complete PDE solver by itself to converge to a solution. Figure 6, shown as a first example, indicates that the results obtained from the models trained with a larger time-step at $t = \{25, 50, 100, 150\}$ are in good agreement with the target data for the difficult uniform-Bunsen case. The flame dynamics are predicted accurately while using less simulation steps. For the uniform-Bunsen case, the hybrid approach takes $8.23 \pm 0.011s$ to infer one simulation run, whereas the complete PDE solver requires $15.21 \pm 0.003s$. Similar performance gains are observed across the other cases studied. Note that for the previous cases, the trained model incurs a negligible runtime overhead. In these cases, even for costly runs with high resolutions the incomplete PDE solver would not converge to the solutions of the complete description.

The last column of table 1 (*Hybrid NN-PDE-dt*) summarizes the errors of the large time-step approach for all four reactive cases considered. Despite an effectively doubled computational performance, this model achieves similar errors to the hybrid NN-PDE approach. This highlights the capabilities of learned, hybrid PDE solvers, which can produce these solutions without the stability problems exhibited by the complete PDE solver, while at the same time being more accurate than pure data-driven predictions.

The PDD approach is not restricted by the time-step of the PDE solver. Therefore, we investigate the performance of the PDD approach on the nonUniform-Bunsen32 case with larger time-steps as

Figure 7: Temperature field predictions achieved by the neural network model $\tilde{T}_*^{NN}$ from given initial flame shape $T_0$ to achieve target flame shape $\tilde{T}_*$ by controlling flow velocity.

shown in the figure 5 (B). PDD achieves higher MAPE of 22.44 % for $8\Delta t_c$ setup as compared to the MAPE of 12.48 % for $\Delta t_c$ setup. The PDD approach does not predict the dynamics accurately for any of the larger time-steps considered. Additional visualizations can be found in appendix C.

## 5.3 FLAME SHAPE CONTROL

We lastly demonstrate the potential of our approach with a problem from multi-physics system control: we test the joint applicability of a neural network with reactive flow solver to obtain a desired flame shape. We use a predictor-corrector approach (Holl et al., 2019) with a differentiable physics loss function and control sequence refinement for nonUniform-Bunsen case with $32 \times 32$ resolution. The learning objective is to arrive at a target temperature field with the desired flame shape from an arbitrary, given initial configuration by controlling the velocity field. We assume that the temperature and mass-fraction fields are observable and only the velocity field is controllable. Figure 7 (a-e) shows the transition achieved and compares the target flame shape obtained using the neural network model with the target flame shape ($T_*$) for a test case. Figure 7 (f,g) compares the flame shape obtained for additional test cases. The neural network model is able to control the velocity of fuel-air mixture flow to obtain the desired flame shape. To quantify these results, the learned model achieves MAPE of $1.87 \pm 0.73\%$ over 50 test cases considered. It achieves an improvement of 52% compared to a baseline model without a predictor network for the long term prediction of temperature fields. This baseline model has a MAPE of $3.91 \pm 2.19\%$. This case highlights the capabilities of differentiable multi-physics solvers, and that our full neural network model is able to reliably control the complex physics of the reactive flow over long time spans.

## 6 CONCLUSIONS

These results demonstrate that neural networks can be employed for learning the completion of partial PDE solvers in the context of a multi-physics system. The trained neural network model works alongside a partial solver to accurately account for and predict the effects of unknown physics. The ability of the presented hybrid NN-PDE approach to predict the long-term temporal evolution of multi-physics systems was demonstrated on the specific case of a reacting flow in different configurations. When compared with a purely data-driven approach and state-of-the-art FNO approach, the qualitative and quantitative performance of the hybrid NN-PDE model is found to be clearly superior. The incomplete PDE description helps the neural network model recover the target simulation with a significantly improved accuracy. This hybrid NN-PDE model is able to predict the correct evolution of the important features of the multi-physics system (in our case, the flame interface and flame shape) for longer simulation steps in all scenarios and is able to generalize to other (initial and boundary) conditions of the system. This is demonstrated by variations in the equivalence ratio and inlet velocity forcing. We also show that such hybrid approach may have the added benefits of allowing to relax numerical constraints linked to the potential stiffness of the unknown physics.

Our work represents a stepping stone for numerous avenues of future work, e.g., the proposed hybrid NN-PDE solver could be further utilized to predict and control the dynamics of other tightly coupled systems (Levy, 1999; Dowell & Hall, 2001). Here, the inclusion of transfer learning techniques would pose an interesting avenue for future work. We note that the proposed approach does not seek to replace the classical physical simulation approaches. However, it is an important step in the direction of harnessing the capabilities of neural network models using the knowledge of partial differential equations for complex multi-physical systems.

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

APPENDIX

## A  NEURAL NETWORK ARCHITECTURE

All results provided in the main paper use a fully convolutional neural network. Different input flow field quantities have different value ranges of the data e.g. $T \in [400, 2500]$, $Y_f \in [0, 0.05]$. Therefore, the input data provided to the neural network model is normalized using mean and standard deviation. The output of the neural network model is then transformed back using the same mean and standard deviation. When predicting correction of $Y_k$ fields, the output of the neural network is constrained to provide values in the range $Y_f \in [0, 0.05]$, $Y_o \in [0, 1]$. In the hybrid NN-PDE approach, the output of the neural network model is provided to the incomplete PDE solver as an input, as shown in figure 2. Each network is trained for 100 epochs with learning rate of 0.0001. An $L_2$ based loss function is used as defined in section 3.3.

We additionally experimented with ResNet and UNet architectures for the neural network part of the purely data driven and hybrid NN-PDE approach discussed in the paper. Table 2 provides details of the convolutional blocks, layers of the network, and activation function. The *UNet32* architecture with 2 layers is used for $32 \times 32$ cases and the *UNet100* architecture with 3 layers is used for $100 \times 100$ case discussed in the paper. Table 3 shows the comparison between MAPE achieved by ResNet vs. UNet architecture for the purely data-driven and the hybrid NN-PDE approach for different cases considered in the paper. For the hybrid NN-PDE approach with $32 \times 32$ resolution cases, ResNet and UNet achieve comparable performance with the UNet yielding the lowest error. The main difference arises in the $100 \times 100$ resolution case. The ResNet fails to converge during training, leading to high test error. On the contrary, the UNet achieves low training and testing errors. The main advantage of UNet comes from transforming high resolution input to low resolution representation and reconstructing it back to the high resolution output. Therefore, we choose the UNet architecture for the hybrid NN-PDE approach. For the purely data-driven approach ResNet was found to be better for 2 out of 3 cases. Specifically for the nonUniform-Bunsen32 case, UNet performs very poorly for the PDD approach. Therefore, we report ResNet errors for the PDD approach throughout the paper. Figure 2 shows the details of the predictor block for PDD and hybrid NN-PDE approach.

Table 2: Details of various neural network architectures used in this paper.

| Hyper-parameters | ResNet | UNet32 | UNet100 |
|---|---|---|---|
| Kernel size | 5 | 5 | 5 |
| Latent size | 32 | | |
| Activation | LeakyReLu | LeakyReLu | ReLu |
| Loss | MSE | MSE | MSE |
| # ResBlocks | 5 | | |
| # UNet layers | | 2 | 3 |
| CNN stack depth | 2 | 3 | 2 |
| Base latent size | | 16 | 16 |
| Spacial down-sample by layer | | 2 | 2 |
| latent sizes | | (32) | (32,64) |
| # trainable parameters | 261,953 | 136,227 | 362,371 |

Table 3: Mean and standard deviation of errors over different architectures for purely data-driven and hybrid NN-PDE approach.

| | | PDD ResNet | PDD UNet | Hybrid NN-PDE ResNet | Hybrid NN-PDE UNet |
|---|---|---|---|---|---|
| MAPE | Planar-v0 | $6.33 \pm 3.05\%$ | $7.09 \pm 4.40\%$ | $1.62 \pm 0.44\%$ | $\mathbf{1.40} \pm 0.65\%$ |
| | uniform-Bunsen | $7.58 \pm 3.73\%$ | $2.87 \pm 1.53\%$ | $2.01 \pm 0.99\%$ | $\mathbf{0.72} \pm 0.37\%$ |
| | nonUniform-Bunsen32 | $12.48 \pm 11.31\%$ | $19.19 \pm 14.92\%$ | $3.25 \pm 2.35\%$ | $\mathbf{2.04} \pm 1.39\%$ |
| | nonUniform-Bunsen100 | - | - | $15.68 \pm 10.44\%$ | $\mathbf{3.23} \pm 3.76\%$ |

Table 4: Details of the boundary conditions used for the planar flame case and various cases of Bunsen-type flames. The uniform-Bunsen case is obtained with $\kappa$ = constant and nonUniform-Bunsen case is obtained with $\kappa \in \mathbb{R}^n$ as discussed in section 4

| Planar flame | | | Bunsen-type flame | | |
|---|---|---|---|---|---|
| field | inlet | left and right wall | field | inlet | left and right wall |
| T | 400 K | Neumann BC | T | 800 K | Neumann BC |
| $u_y$ | 0 | - | $u_y$ | $\kappa$ | no-slip |
| $u_x$ | 0 | - | $u_x$ | 0 | slip |
| P | 101325 | Neumann BC | P | 101325 | Neumann BC |
| $Y_f$ | $\frac{1}{1+\frac{\varphi}{E}(1+3.76\frac{W_{N_2}}{W_{O_2}})}$ | slip | $Y_f$ | $\frac{1}{1+\frac{\varphi}{E}(1+3.76\frac{W_{N_2}}{W_{O_2}})}$ | slip |

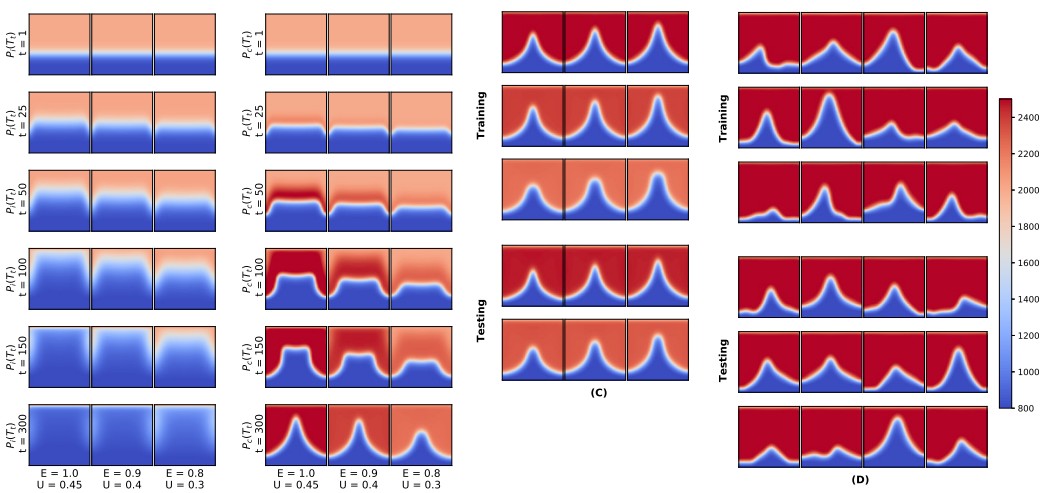

Figure 8: Instances of (A): incomplete PDE solver; (B): complete PDE solver for different operating conditions. (C): Snapshot of complete PDE solver at $t = 300$ for all training and testing datasets of uniform-Bunsen case. (D): Snapshot of complete PDE solver at $t = 500$ for all training and testing datasets of nonUniform-Bunsen32 case.

# B  DATA GENERATION

To generate input and target data for training, we simulate the temperature $T$, mass fractions $Y_f$, $Y_o$ and velocity $u_x, u_y$ fields of all flames under study. Table 4 summarizes the boundary conditions applied for the planar and Bunsen-type flame cases discussed in the paper. No-slip boundary conditions are used for the velocity field $u_y$ to obtain the $\Lambda$-shaped flames.

We consider variations in equivalence ratio $E$ and inlet velocity amplitude $u_y$ to obtain training and test datasets. Figure 8 (A) shows the temperature field evolution of the incomplete PDE solver at different time instances for 3 different operating conditions for the uniform-Bunsen case. Figure 8 (B) shows the corresponding target training data for the uniform-Bunsen case. All the simulations are run for 300 steps. It shows the difference between non-reactive and reactive flow solver simulations for different operating conditions over 300 time-steps. Figure 8 (C) shows the last snapshot ($t = 300$) of 9 training simulations and 6 testing datasets with operating parameters interpolated between the training data parameters for uniform-Bunsen case. These datasets are obtained by varying equivalence ratio and constant inlet velocity as discussed in section 4. Flame temperature depends on the equivalence ratio used and flame height depends on the inlet velocity amplitude. Figure 8 (D) showcases the snapshots of training and testing dataset at $t = 500$ for nonUniform-Bunsen32 case. Non-uniform variations in inlet velocity profile leads to different complex flame shapes. We use 12 datasets with 500 simulation steps to train the models and test it on 12 test cases shown in figure 8 (D).

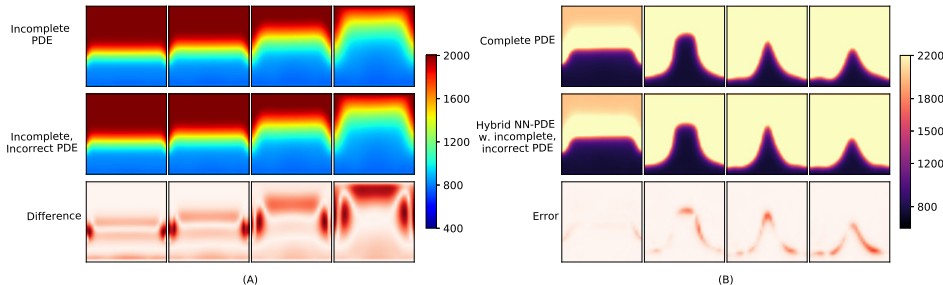

Figure 9: Generalization to incorrect PDE parameters (A) visualization of differences in temperature fields due to incorrect parameters in incomplete PDE description. (B) Modified hybrid NN-PDE model, which combines the incomplete, incorrect PDE solver with neural network model is able to recover solutions of complete, correct PDE.

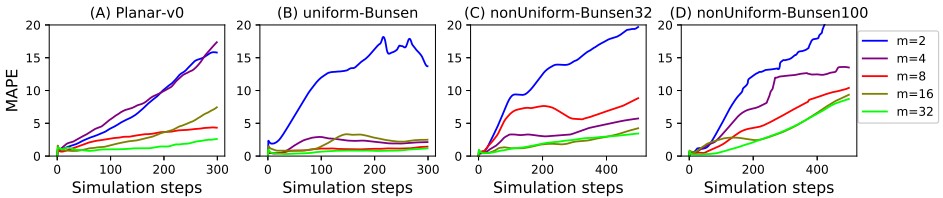

Figure 10: Effect of longer look-ahead steps. MAPE of temperature field predictions by hybrid NN-PDE model, over all testsets. Models trained with higher look-ahead steps $m$ accurately predict the temporal evolution of dynamics for longer duration across all cases considered.

## C  ABLATION STUDY

**Effect of training dataset size** We study the effect of training dataset size on the accuracy of PDD predictions. We train PDD models with different numbers of simulations in the training dataset. Each simulation contains 500 time-steps. Figure 5 (A) shows the MAPE of PDD models trained with $\{4, 8, 12, 16, 24, 32\}$ training datasets, over a fixed testset. We compare the performance of these PDD models with an equivalent (trained with same look-ahead steps $m = 2$) hybrid NN-PDE model, trained with 12 simulation sets. Figure 5 (A) shows increasing the number of training sets has little or no effect on the prediction capabilities of the PDD models for nonUniform-Bunsen32 case. The hybrid model with 12 training sets achieves a MAPE of $12.49 \pm 4.17\%$, an improvement of 38% over the PDD model with 32 training sets. The purely data-driven models cannot achieve the same level of accuracy even in the presence of large amounts of data.

**Generalization to incorrect PDE parameters** The proposed hybrid NN-PDE model is capable of completing the PDE description even if the underlying incomplete PDE solver has incorrect parameters. We refer to the incomplete solver with incorrect parameters as 'incomplete, incorrect PDE'. We experiment with a modified hybrid NN-PDE approach wherein we combine an incomplete, incorrect PDE solver with a neural network model. We assume that the known values of the incomplete PDE parameters in equation 2, strain rate tensor ($\tau$), the diffusion coefficient of species $k$ ($D_k$) and mixture thermal conductivity ($\lambda$), are incorrect. Figure 9 (A) shows the difference between temperature field evolution of the incomplete PDE solver with correct parameters ($[\tau, D_k, \lambda] = [0.1, 0.1, 0.1]$) and incorrect parameters ($[\tau, D_k, \lambda] = [0.05, 0.05, 0.05]$) at different time-steps. The hybrid NN-PDE combines this incomplete, incorrect PDE solver with the neural network model to obtain the solutions of the complete PDE solver with correct parameters. Figure 9 (B) compares the flame dynamics predicted by this hybrid NN-PDE model with that of the complete, correct PDE solver. A good match is observed over various test cases. The hybrid model with incomplete, incorrect solver achieves an overall MAPE of $2.48 \pm 1.20$ %, compared to the MAPE of $2.04 \pm 1.39$ % for hybrid model with incomplete, correct PDE.

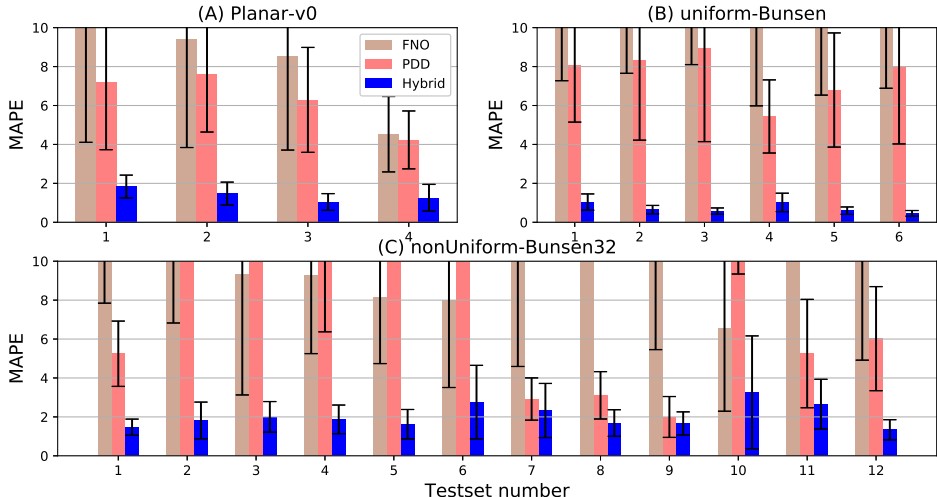

Figure 11: Bar plot of MAPE of temperature field predictions by FNO, PDD and hybrid NN-PDE model at time-step $\Delta t_c$, for different testcases.

**Effect of longer look-ahead steps** We evaluate the effect of varying look-ahead steps on the performance of the hybrid model. We show the comparison of models trained with $m = \{2, 4, 8, 16, 32\}$ for different cases in figure 10. Models with smaller $m$ (2) do not learn to accurately correct the fields over long time and quickly diverge from the target simulation. Using larger $m$ improves the quality of prediction drastically as the model learns the correction via the gradients over longer simulation steps. For Planar-v0 and uniform-Bunsen cases, iterating the NN and PDE solver for 32 time-steps improves the accuracy by 81.7% and 94% respectively compared to the $m = 2$ model. For nonUniform-Bunsen32 and nonUniform-Bunsen100, the model performance improves by 84.2% and 70.9% respectively, by using $m = 32$ instead of $m = 2$ model.

**Effect of temporal coarsening** Figure 11 and table 1 compare the MAPE of temperature field predictions at time-step $\Delta t_c$, for all three scenarios. As seen from the figure 11, the hybrid NN-PDE approach consistently performs better than both the baselines for all three scenarios considered. Higher standard deviations in Table 1 indicate the large differences in the prediction of flame temperatures for baseline approaches. The multi-physics systems we study provide a substantially more difficult environment than regular fluids: the chemical reactions are numerically very stiff, and the resulting dynamic interactions are difficult to capture by numerical solvers. To further illustrate the temporal behavior of the simulations, we perform additional experiments with the baseline approaches using twice the time-step size of the complete solver ($2\Delta t_c$). As shown in figure 12, for the Planar-v0 and uniform-Bunsen case, PDD catches up with the hybrid approach whereas the PDD approach completely fails to predict the dynamics of the nonUniform-Bunsen32 case. We further investigate the nonUniform-Bunsen32 case with larger time-steps (2 times, 4 times and 8 times) as shown in figure 5 (B). It shows MAPE of the predictions over all the testsets and simulation steps by the PDD model. The PDD approach does not predict the dynamics accurately for any of the larger time-steps considered. Figure 13 visualize these results over different time-steps for one of the test cases. The predictions made by FNO, PDD and hybrid NN-PDE model for nonUniform-Bunsen32 case at twice the time-step of $2\Delta t_c$, are compared with the ground truth solutions coming from the complete PDE solver at time-step of $\Delta t_c$. FNO and PDD (upper two rows) fail to recover the correct flame shapes over 250 simulation steps. The hybrid NN-PDE model (second row from below) predicts the flame shape accurately, thus relaxing the temporal stiffness of the complete PDE solver.

# D   ADDITIONAL RESULTS

In this section, we present additional visualizations of all reactive flow scenarios considered in section 4 for different test cases. Additionally, we discuss the results of the mass fraction fields we obtained. Table 5 summarizes MSE values of fuel ($Y_f$) and oxidizer ($Y_o$) mass fraction fields for

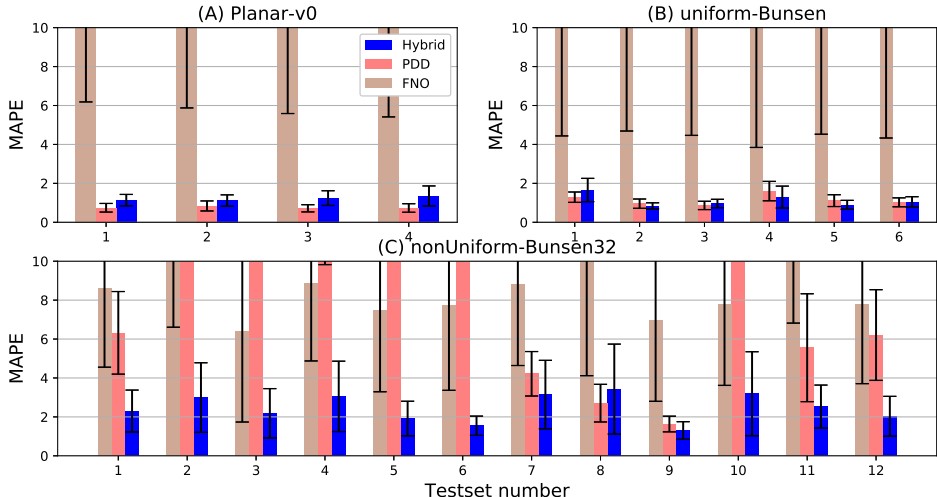

Figure 12: Bar plot of MAPE of temperature field predictions by FNO, PDD and hybrid NN-PDE model at time-step $2\Delta t_c$, for different testcases.

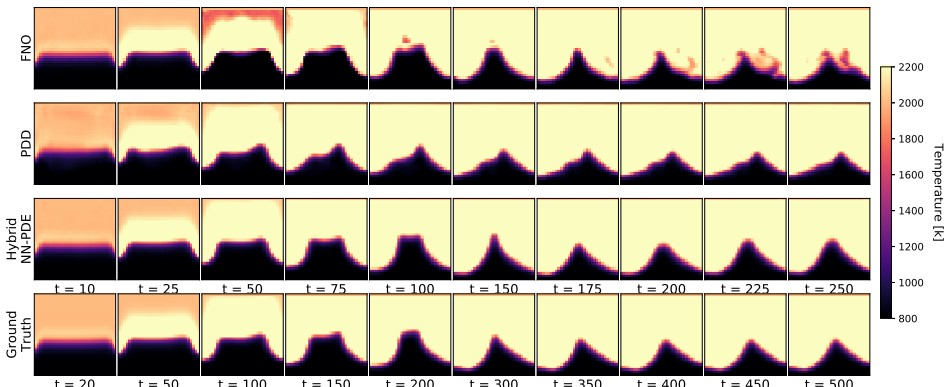

Figure 13: Comparison between different approaches at time-step $2\Delta t_c$ - from top to bottom - FNO, PDD, hybrid NN-PDE for nonUniform-Bunsen32 test case. Hybrid approach predictions accurately match with the ground truth data over long roll-outs.

different baselines and the hybrid NN-PDE approach. We observe similar performance to the temperature field predictions. The hybrid NN-PDE model consistently outperforms the FNO and PDD approaches for all the scenarios considered. The hybrid NN-PDE model is able to simultaneously correct temperature and mass fraction fields, thus successfully modeling the effects of unknown chemical reactions on all the fields of the flow state.

## D.1 PLANAR-V0

Figure 14(A) shows the 2D visualization of the temperature field predictions for Planar-v0 case. As seen from the ground truth images, methane-air mixture (black color) converts into the burned products (yellow color) due the chemical reaction at the flat flame surface (red color). The dotted, horizontal red line helps to compare the transition of the flame interface, i.e. the displacement of the flame front. Due to the chemical reaction, fuel-air mixture is consumed and turns into burned product as the simulation progresses. The FNO approach completely fails to predict the propagation of the flame for the given test case. Its output does not show any evolution from the initial temperature profile for the given operating condition. The purely data-driven approach fails to capture the flame front displacement correctly, thus leading to an inaccurate prediction with large errors. The

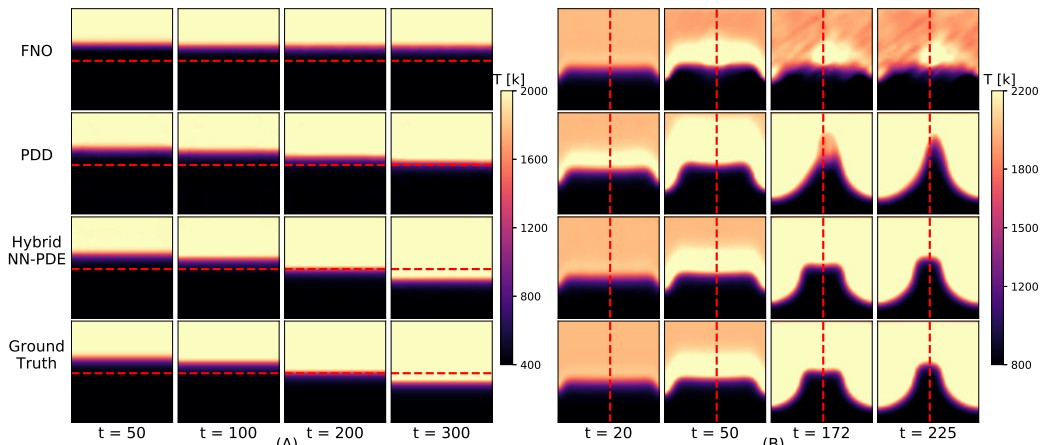

Figure 14: Temperature field predictions for (A): Planar-v0 flame case with $E = 0.95$; (B) uniform-Bunsen case with constant inlet velocity $U = 0.375$ and $E = 0.95$ for different approaches. Hybrid NN-PDE model predicts physically accurate evolution of the flame cases under study.

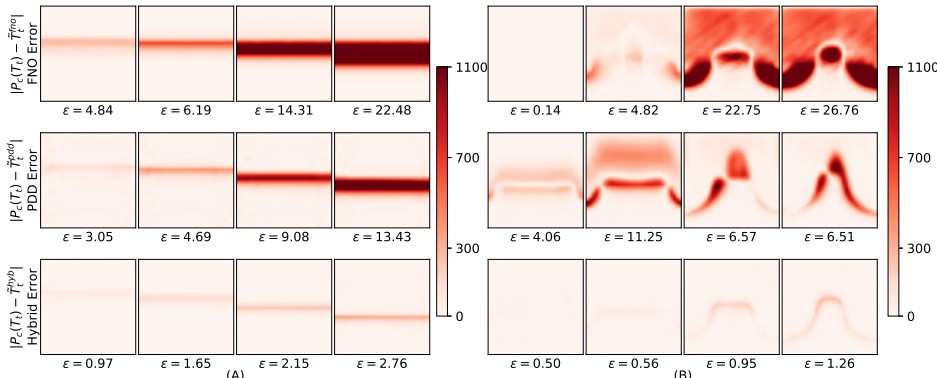

Figure 15: Absolute error between target field ($\mathcal{P}_c(T_t)$) and the output predicted - from top to bottom top - by: FNO ($\tilde{T}_t^{fno}$), purely data-driven approach ($\tilde{T}_t^{pdd}$) and hybrid NN-PDE ($\tilde{T}_t^{hyb}$) model, for the instances shown in figure 14. The numbers represent instantaneous MAPE.

Table 5: Mean and standard deviation of squared errors of fuel ($Y_f$) and oxidizer ($Y_o$) mass fraction fields over all time steps of all testsets. The Hybrid NN-PDE approach outperforms all other baselines considered.

| | | Baseline FNO | Baseline NN | Hybrid NN-PDE | Hybrid NN-PDE-dt |
|---|---|---|---|---|---|
| $Y_f$ | Planar-v0 | 1.3e-4 ± 7.4e-5 | 2.9e-5 ± 3.8e-5 | 8.9e-7 ± 1.0e-6 | **1.6e-7** ± 1.5e-7 |
| | uniform-Bunsen | 6.4e-4 ± 2.4e-4 | 5.1e-5 ± 4.1e-5 | **5.6e-7** ± 7.4e-7 | 9.8e-7 ± 1.3e-6 |
| | nonUniform-Bunsen32 | 2.2e-4 ± 1.9e-4 | 9.6e-5 ± 1.0e-4 | **8.0e-6** ± 1.1e-5 | 1.4e-5 ± 1.9e-5 |
| | nonUniform-Bunsen100 | - | - | **1.8e-5** ± 2.8e-5 | 1.8e-5 ± 2.9e-5 |
| $Y_o$ | Planar-v0 | 1.0e-2 ± 6.8e-3 | 1.3e-3 ± 1.2e-3 | **3.7e-5** ± 2.9e-5 | 4.1e-5 ± 4.4e-5 |
| | uniform-Bunsen | 4.7e-1 ± 9.6e-2 | 2.1e-2 ± 2.7e-2 | **3.6e-5** ± 3.0e-5 | 1.4e-4 ± 1.2e-4 |
| | nonUniform-Bunsen32 | 1.4e0 ± 1.8e0 | 6.1e-3 ± 7.9e-3 | **1.6e-4** ± 1.6e-4 | 2.8e-4 ± 3.0e-4 |
| | nonUniform-Bunsen100 | - | - | **3.3e-4** ± 4.2e-4 | 5.0e-4 ± 5.8e-4 |

hybrid NN-PDE model accurately captures this evolution of planar methane-air flame in a quiescent mixture. Figure 15 shows the instantaneous MAPE w.r.t. ground truth data for predictions shown in figure 14. The absolute error shown in figure 15(A) exceeds 1100 K for the FNO and purely data-driven approaches as these do not predict the flame temperature correctly. We use the upper limit of 1100 K for colorbar to highlight the errors in the hybrid NN-PDE approach more clearly. Large errors in FNO and PDD results stem from their inability to reliably predict the flame temperature and flame front displacement.

### D.2 UNIFORM-BUNSEN

Figures 14(B) and 15(B) show an example of uniform-Bunsen case; a test case with $u_y = 0.375$ and $E = 0.95$. The constant inlet velocity results in a symmetric, Λ-shaped flame. Vertical, dotted, red line in figure 14(B) helps to assess the symmetric nature of the flame. The purely data-driven model predicts a thicker flame ($t = 50$) or a flame with a spurious tip ($t = 172$) or an asymmetric flame ($t = 225$). Snapshots of the hybrid NN-PDE model predictions in figure 14(B) show that it adapts to this scenario very well and succeeds in obtaining the correct results for long term forecasts of the temperature field. Furthermore, the flame shape and height are also better predicted, as shown in figure 14(B). Very low error levels in figure 15(B), such as $\epsilon = 1.26$ at $t = 300$, indicate that the hybrid model recovers the temperature field satisfactorily. Figure 16 compares the performance of the hybrid NN-PDE model for the $Y_f$ and $Y_o$ fields for two different test cases. We again observe that the hybrid NN-PDE model predicts all the fields accurately with very low MSE.

### D.3 NONUNIFORM-BUNSEN

Figure 4 compares the temporal predictions made by FNO, PDD and the hybrid NN-PDE model with ground truth data. Compared to Planar-v0 and uniform-Bunsen case, FNO predicts qualitatively better results for this case. Although highly still accurate, it predicts shapes that come closer to the target flame shapes for the later part of the simulation (for $t > 300$). This improvement might be due to the larger training dataset used for this scenario compared to the previous two scenarios. However, it fails to predict accurate temporal predictions over the entire simulation. The hybrid NN-PDE approach satisfactorily predicts the accurate flame evolution. Additionally, we showcase the performance of the hybrid NN-PDE approach on two different test cases with complex flame shapes in figure 18. The neural network model along with incomplete PDE solver reconstructs these complex flame shapes in an accurate manner.

Finally we showcase the performance of the hybrid NN-PDE approach on different test cases for highly resolved flame case of nonUniform-Bunsen100. Figure 19 compares the predictions made by hybrid NN-PDE over $\{0, 100, 200, 250, 300, 325, 350, 400, 450, 500\}$ simulation steps with the ground truth data. It also shows the absolute difference between them. As the simulation progresses, higher errors are observed around the flame front. However, the hybrid NN-PDE approach captures the flame shape very accurately for longer roll-outs of 500 simulation steps. Currently this approach uses training dataset similar to nonUniform-Bunsen32. Further improvements in accuracy can be

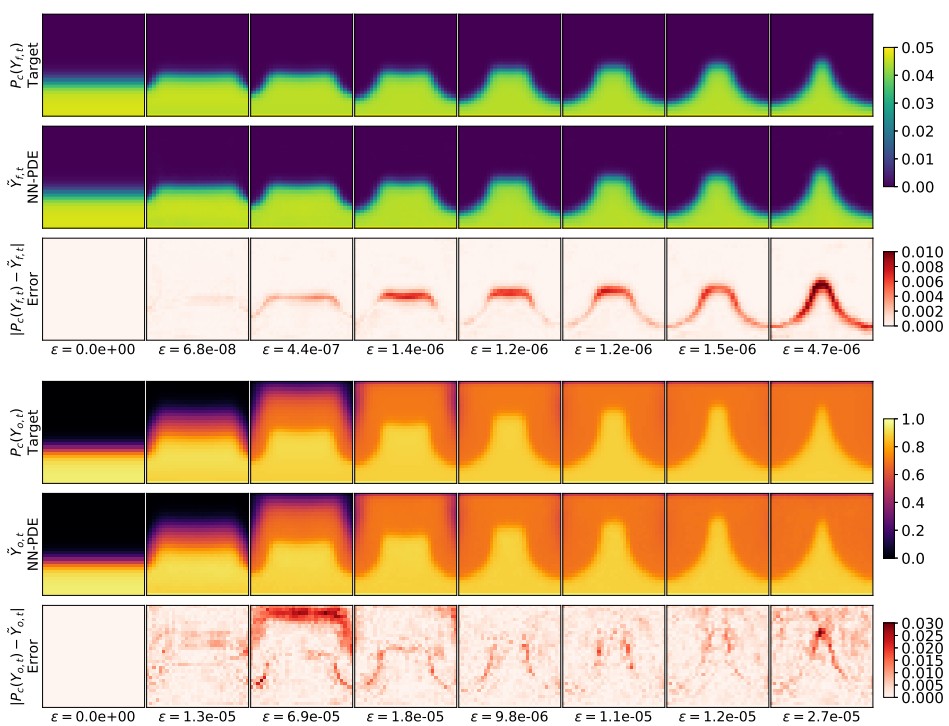

Figure 16: Comparison between ground truth (or Target) data and the hybrid NN-PDE approach for the Uniform-Bunsen case for fuel (top) and oxidizer (bottom) mass fraction fields for different operating conditions. Error field shows the absolute difference between target state and hybrid prediction over $\{0, 50, 100, 150, 200, 225, 250, 300\}$ steps. $\epsilon$ indicates the instantaneous MSE.

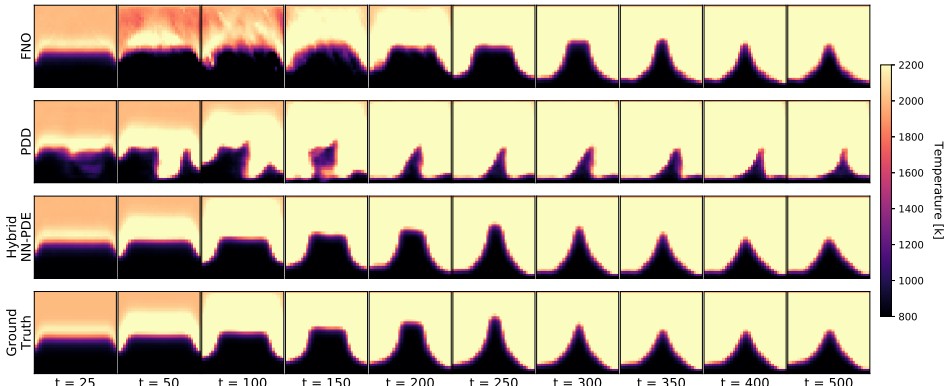

Figure 17: Comparison between different approaches - from top to bottom - FNO, PDD, hybrid NN-PDE for nonUniform-Bunsen32 test case. Hybrid approach predictions accurately match with the ground truth data over long-rollouts of 500 time steps.

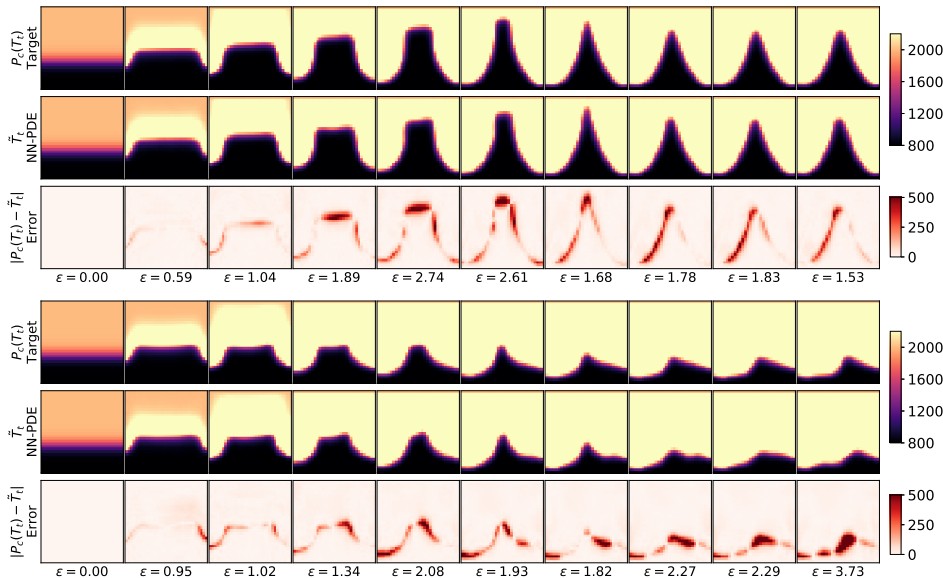

Figure 18: Additional visualization of reactive flow trajectories predicted by the hybrid approach for different test cases of nonUniform-Bunsen32 scenario.

achieved by increasing the training dataset size or training the hybrid model with longer look-ahead steps.

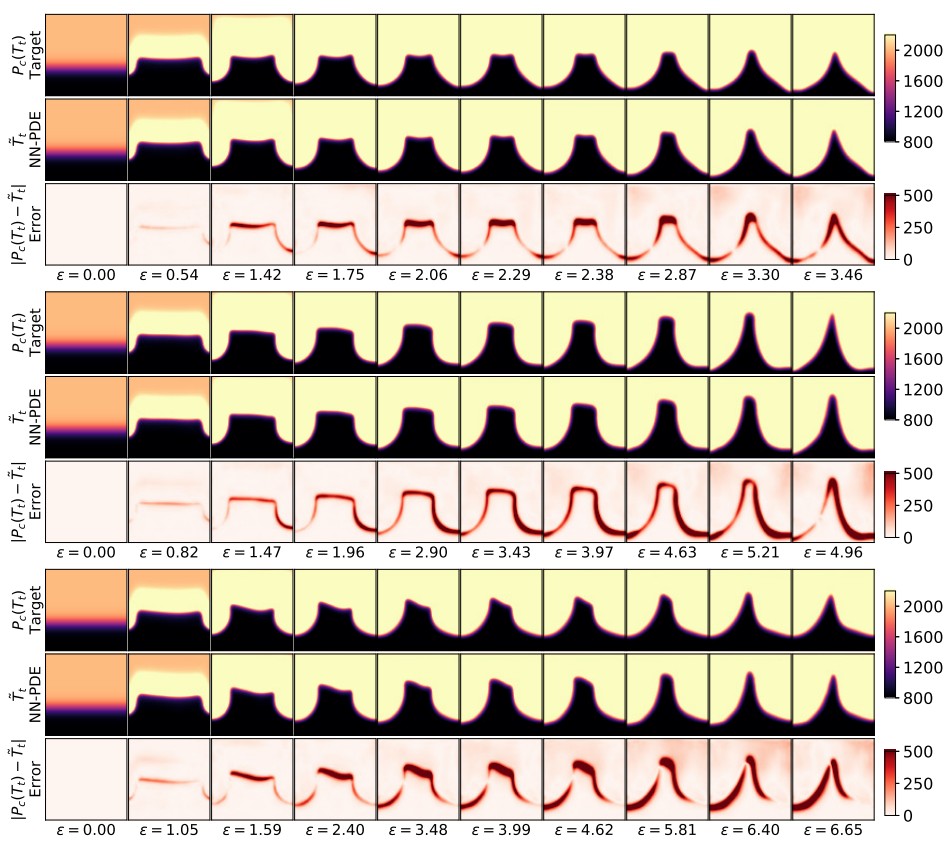

Figure 19: Comparison between ground truth data ($\mathcal{P}_c(T_t)$) and hybrid NN-PDE approach predictions ($\tilde{T}_t$) for various test cases of complex, high resolution scenario of nonUniform-Bunsen100

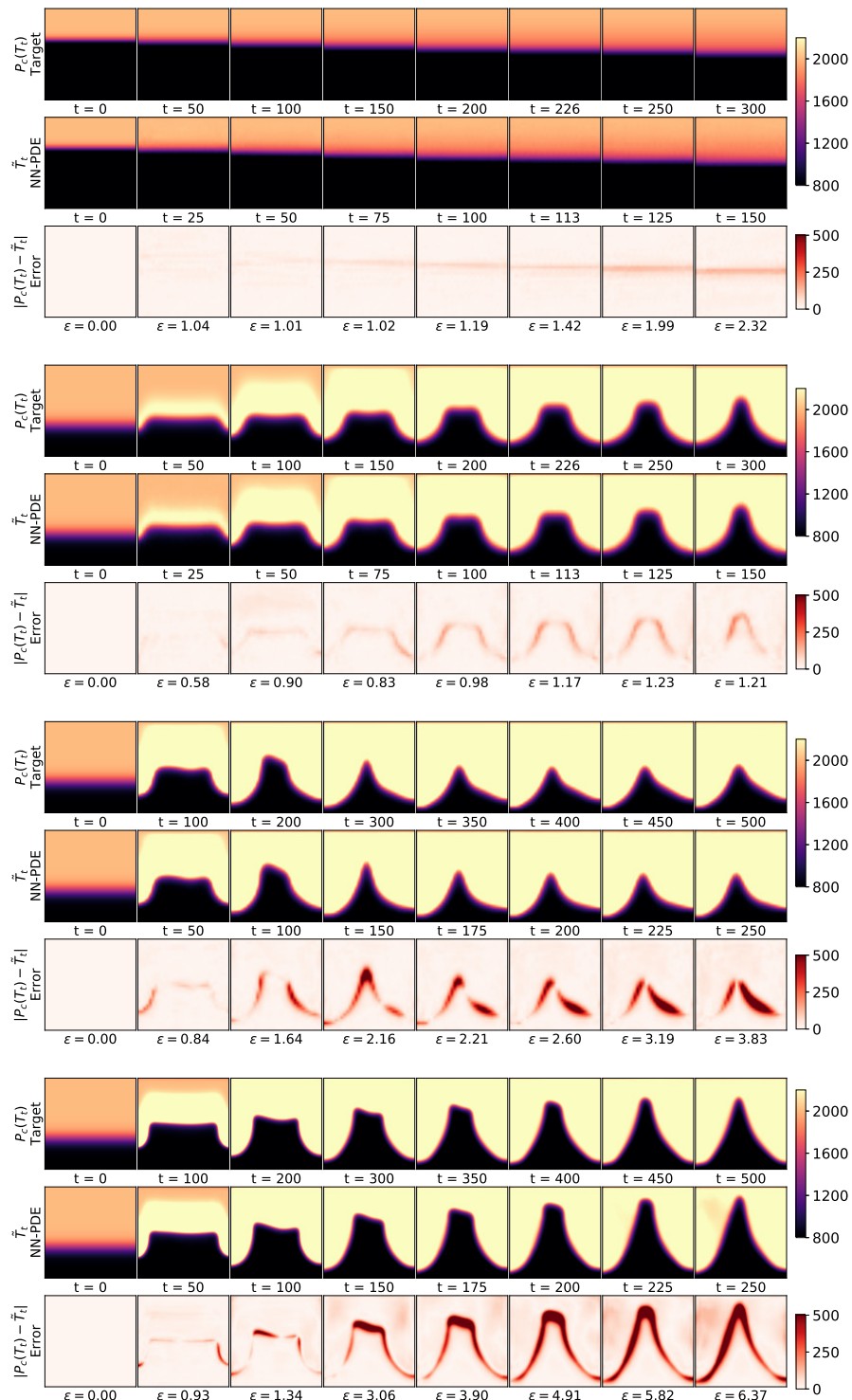

Figure 20: Predictions made by hybrid NN-PDE model ($\tilde{T}_t$) with an incomplete PDE solver at twice the time-step of $2\Delta t_c$, are compared with the ground truth solutions coming from the complete PDE solver at time-step of $\Delta t_c$. We showcase the effectiveness of hybrid approach in relaxing temporal stiffness of the complete PDE solver on reactive flow cases of - from top to bottom - Planar-v0, uniform-Bunsen, nonUniform-Bunsen32 and nonUniform-Bunsen100 for different test cases.

