# OpenReview forum: "Incomplete to complete multiphysics forecasting - a hybrid approach for learning unknown phenomena"
_ICLR.cc/2023/Conference — Submitted to ICLR 2023_

### Official Review · Reviewer_NRLv · 2022-10-23

**Confidence:** 3
**Correctness:** 2
**Technical Novelty And Significance:** 2
**Empirical Novelty And Significance:** 2
**Recommendation:** 3

**Clarity, Quality, Novelty And Reproducibility:**

The paper is mostly easy to follow. I have some doubts with respect to the originality since it seems to me that APHYNITY and Physics VAE address very similar, of not the same settings.

**Strength And Weaknesses:**

Weaknesses

The paper claims that it is the first of each kind that learns to complete physics in a PDE setting. Nevertheless learning with incomplete physics models using neural networks to complete the missing part has been studied rather extensively. For example: APHYNITY, Yin et a, Journal of Statistical Mechanics: Theory and Experiment (also cited in beginning of the present submission)  and ICLR2021; Physics integrated VAEs for robust and interpretable generative modelling, Takeishi & Kalousis, NeurIPS, 2021. APHYNITY tackles forecasting tasks, similar to the current submission, for physics systems that are described by ODES/PDES. It models the underlying dynamical system with two additive components, one given by the known physics and a neural network modelling the unknown parts. Physics VAE learns generative models where the VAE decoder also decomposes, to a more general composition than additive, of an incomplete physics component and a neural network. The paper should at least discuss differences from these two works.

With respect to the experimental evaluation it is not clear to me how the neural network baseline has been trained. It seems that it learns state-state transitions. How is this done? does the model during training sees only consecutive state pairs? or does the model evolve in a longer horizon, i.e. over a sequence of states, and then the model update would have to happen over such sequences, accounting like that for longer temporal dependencies and error compounding. From what I understand the neural network baseline is trained over state state transitions, which, if I am correct, puts it at a clear disadavantage with the hybrid model. More appropriate baselines will explicitly model for long term dependencies, e.g. a solver where there is no physics but just a neural network, neural ODEs, models with recurency. The two papers above provide a number of such baselines.

Strengths
There is a detailed evaluation of the model in a complex scenario exploring different extrapolation scenarios.


**Summary Of The Paper:**

The paper considers the case of incomplete physics for numerically solving Partial Differential Equations. It uses a neural network to complete the missing terms and more accurately forecast the evolution of the dynamical system. The proposed approach is evaluated on different variants of a reactive flow problem governed by Navier-Stokes equations.

**Summary Of The Review:**

This is mostly an application paper that proposes to learn to complete incomplete physics models using neural networks in the context of reactive flow problems. My main concerns with the paper is how it relates to previous work that also learns to complete incomplete physics models as well as the fact that, at least to my understanding, the baselines that are considered are weak.

---

> ### Author Response · Authors · 2022-11-18
> **Response to Reviewer NRLv**
>
> **Discussion on related works:** Thank you for your comments. We apologize for not including these works in our previous manuscript. We have updated the section ‘Related work’ to discuss these papers. Yin et al. 2021 (APHYNITY) and Takeishi and Kalousis 2021 (Physics integrated VAEs) introduce frameworks of augmenting incomplete physical dynamics with neural network models to forecast the solutions of a complete dynamical system. These approaches demonstrate the applicability on ODE/PDE systems, which are weakly non-linear and the unknown dynamics are linear combinations of the underlying flow fields. We explore a significantly more challenging application: a reactive flow. Note that a reactive flow is characterized by a multi-physics system with non-linear advective terms and strongly nonlinear dynamics, described by exponential source terms that exhibit non-linear combinations of the flow fields. Our hybrid approach is able to predict the dynamics accurately over very long rollouts of 500 steps, compared to the short rollouts of 25 or 50 steps demonstrated with APHYNITY and Physics integrated VAEs, respectively.
>
> As such, we believe our work represents an important continuation of the line of work started by these two papers.
>
> **With respect to the experimental evaluation it is not clear to me how the neural network baseline has been trained.**
>
> We would like to clarify that the baselines were not put at a disadvantage: the training of the purely data-driven (PDD) approach and the Fourier neural operator versions indeed include longer temporal dependencies. As shown in figure 2(A) of the manuscript, a multi-step training framework was used to account for longer temporal dependencies and the compounding of errors. Via this setup, we can very reliably conclude from a large number of training runs that the improved performance of the hybrid model really stems from jointly training with the physics solver, rather than artificial disadvantages imposed by the learning setup.
>
> More specifically, the predictor block in figure 2 (A) consists of only the neural network model in case of purely data-driven approach, figure 2 (B1), the Fourier neural operator model from Li et al. (2020a) in case of FNO baseline, and the incomplete PDE solver with neural network model in case of hybrid NN-PDE model, as shown figure 2 (B2). Thus, the PDD and FNO models are treated in exactly the same manner as hybrid NN-PDE model in terms of recurrency used, denoted by the look-ahead steps $m$ in the paper. The same can be seen from figure 2 and loss function formulation in equation 5. All the models compared in table 1 are trained with 32 look-ahead steps, including PDD and FNO baselines. We have added the following comment in section 5 of the main manuscript to clarify the setup of the baselines: “As baselines, we compare against a purely data-driven approach; a neural network model with exactly the same look-ahead steps, and the Fourier neural operator of Li et al. (2020a) that likewise includes $m$ look-ahead steps as discussed in section 3.3.”
>
> Thus, PDD or FNO had no disadvantage from the training setup used. The performance gains achieved by hybrid NN-PDE approach come from the integration of the incomplete PDE solver.

---

### Official Review · Reviewer_95Sg · 2022-10-24

**Confidence:** 5
**Correctness:** 3
**Technical Novelty And Significance:** 3
**Empirical Novelty And Significance:** 3
**Recommendation:** 6

**Clarity, Quality, Novelty And Reproducibility:**

The paper is clearly written and provides a decent account for the choices made to set up the test problem. The use of existing framework for fluid dynamics solver (phiflow) is a plus. I did not find any reference to the source code/datasets; and given the complexity of the setup (numerical PDE solver + ML model) I think it’s quite hard to independently reproduce the results based on the paper alone.

**Details Of Ethics Concerns:**

I enjoyed reading the paper in general, as the authors address an important question of whether physical priors help/needed to model continuous dynamical systems well. I find the step up in complexity of the simulated system a bonus and in general think this is an important research avenue.

I’m generally leaning towards accepting this paper (score of 8), assuming some flaws and weaknesses can be addressed. (hence the initial score of 6).

My (1) primary concern is related to the evaluation metric and error analysis. Unless I’ve missed something, it is not clear how significant the mean values of MAPE and MSE are given the standard deviations. Is there a way to present the results in a way that clearly indicates that hybrid approach generally improves over data-driven methods? Maybe using a histogram of errors over different sample trajectories / random seeds would provide a more complete picture of error distribution? The fear is that coarse summary might hide stability problems of the purely data driven approaches which would misrepresent their performance as there are numerous methods to address stability issues.

My (2) concern is related to the time-step used for data-driven predictions. In general, data-driven approaches often subsample the predictions in time, which result in better accuracy and efficiency. (e.g. see Fig. 3 “Stachenfeld et al., 2022” (reference used in the paper)). Given that the underlying ground truth solver involves stiff dynamics it could turn out that data-driven models are evaluated in a highly suboptimal regime where they are tasked to predict a very slowly varying solution.

Finally, I think the paper would read better if the stakes for “primacy” were omitted. Arguably a number of previous works learned various closure models in a tandem with differentiable solvers and present work just extends it to a more challenging setting, which is important in its own right.

I’d be happy to provide an additional set of minor comments regarding potential improvements wrt references after primary concerns are addressed.


**Strength And Weaknesses:**

Strengths:
* Authors consider a dynamical system that is more complex than that commonly studied in learned simulators literature.
* The considered method is tested against relevant baseline approaches.
* Paper is clearly written.

Weaknesses:
* Standard deviation estimates of the quantitative results (e.g. Table 1) are O(1) wrt. relevant mean values. While visualizations support the conclusions based on the trend of the mean MAPE and MSE, I find the results somewhat hard to trust.
* An important time subsampling hyperparameter has not been considered for data-driven models.
* AFAICT datasets are not made available to perform independent evaluations/tests.


**Summary Of The Paper:**

The paper studies the task of learning continuous dynamical systems from data using a hybrid (machine learning + PDE solvers) approach and compares the performance to purely data-driven methods. In particular, authors consider a regularly occurring in practice setting where only a partial knowledge of the governing equations are available. Authors use reactive flow systems (specifically various flame simulations) as the test ground for evaluation and comparison. They find that by incorporating physical priors in the form of a partial PDE solver improves the performance of the learned model and is able to accurately capture the dynamics of the system and works well for test simulation settings. Additional findings include observation that learned hybrid models result in less stiff updates compared to the underlying ground truth solver.

**Summary Of The Review:**

I enjoyed reading the paper in general, as the authors address an important question of whether physical priors help/needed to model continuous dynamical systems well. I find the step up in complexity of the simulated system a bonus and in general think this is an important research avenue.

I’m generally leaning towards accepting this paper (score of 8), assuming some flaws and weaknesses can be addressed. (hence the initial score of 6).

My (1) primary concern is related to the evaluation metric and error analysis. Unless I’ve missed something, it is not clear how significant the mean values of MAPE and MSE are given the standard deviations. Is there a way to present the results in a way that clearly indicates that hybrid approach generally improves over data-driven methods? Maybe using a histogram of errors over different sample trajectories / random seeds would provide a more complete picture of error distribution? The fear is that coarse summary might hide stability problems of the purely data driven approaches which would misrepresent their performance as there are numerous methods to address stability issues.

My (2) concern is related to the time-step used for data-driven predictions. In general, data-driven approaches often subsample the predictions in time, which result in better accuracy and efficiency. (e.g. see Fig. 3 “Stachenfeld et al., 2022” (reference used in the paper)). Given that the underlying ground truth solver involves stiff dynamics it could turn out that data-driven models are evaluated in a highly suboptimal regime where they are tasked to predict a very slowly varying solution.

Finally, I think the paper would read better if the stakes for “primacy” were omitted. Arguably a number of previous works learned various closure models in a tandem with differentiable solvers and present work just extends it to a more challenging setting, which is important in its own right.

I’d be happy to provide an additional set of minor comments regarding potential improvements wrt references after primary concerns are addressed.

---

> ### Author Response · Authors · 2022-11-18
> **Response to Reviewer 95Sg**
>
> **While visualizations support the conclusions based on the trend of the mean MAPE and MSE, I find the results somewhat hard to trust.**
>
> In our opinion, the MAPE and MSE metrics clearly show the advantage of the hybrid approach over baselines considered. Flame interface areas have very high temperatures (~ 2000 K). Therefore, any discrepancy in the flame temperature predictions lead to high mean and standard deviation of error values. We have updated our manuscript to add bar graphs of the MAPE over different test cases for all three scenarios. As seen from figure 11, the hybrid NN-PDE approach consistently performs better than both the baselines for all three scenarios considered. We would be happy to include any additional metric to improve the clarity of the results presented.
>
> **An important time subsampling hyperparameter has not been considered for data-driven models.**
>
> We would like to point out that the multi-physics systems we study provide a substantially more difficult environment than regular fluids: the chemical reactions are numerically very stiff, and the resulting dynamic interactions are difficult to capture by numerical solvers. To further illustrate the temporal behavior of the simulations, we performed additional experiments with the baseline approaches using twice the time-step size of the complete solver. Bar plots in figure 12 compare the performance of the baseline approaches across different test cases, with hybrid NN-PDE model, for all three scenarios.  For the Planar-v0 and uniform-Bunsen case, PDD catches up with the hybrid approach whereas the PDD approach completely fails to predict the dynamics of the nonUniform-Bunsen32 case. We further investigate the nonUniform-Bunsen32 case with larger time-steps (2 times, 4 times and 8 times) as shown in figure 5 (B). PDD achieves MAPE of 22.44 ± 16.30 % for $8 \Delta t_c$ setup as compared to the MAPE of 12.48 ± 10.41 % for $\Delta t_c$ setup. The PDD approach does not predict the dynamics accurately for any of the larger time-steps considered.
>
> **I did not find any reference to the source code/datasets**
>
> As mentioned in Section 5 (1st paragraph) of our paper, all source code and datasets that we used will be made publicly available upon acceptance of the publication. We strongly believe in reproducible research, and we will ensure that the published results can be reproduced.
>
>
> **Finally, I think the paper would read better if the stakes for “primacy” were omitted.**
>
> Thank you for the suggestion. We have updated sentences in the introduction section to improve this aspect.

---

### Official Review · Reviewer_hgQY · 2022-11-01

**Confidence:** 3
**Correctness:** 4
**Technical Novelty And Significance:** 3
**Empirical Novelty And Significance:** Not applicable
**Recommendation:** 8

**Clarity, Quality, Novelty And Reproducibility:**

- The paper is very clear.
- The writing style is appropriate and engaging.
- The method presented in the manuscript is understandable also for people which are not familiar with the specific application.
- While I am not domain expert, to my knowledge there are no other references where a NN is trained to correct for some lack of information in the PDE description.
- The method is described in a clear and accessible way making it appealing and deployable to different other downstream tasks involving complicated PDEs.
- The authors assert the code will be released upon acceptance. However, the manuscript provides quite a high level of details which I imagine  would make the results easily reproducible.

**Strength And Weaknesses:**

The problem tackled by the paper is of great practical relevance. It is common in the physical sciences to face numerical PDEs where the complete description is either very expensive, from a computational perspective, or non accessible. Therefore, when a full analytic description is available, as well as reference, experimental, data, one can combine a numerical PDE solver for an incomplete description  of the full PDE with a correction term which can be learned by a NN upon training on some appropriate data.

On the other hand, physics-informed ML is a quite well-established avenue of research. It is not at all surprising to me that combining incomplete physical information of an incomplete PDE description with a NN that learns phenomena acting on different time scales enhance the performance. Therefore, while I am not too impressed nor surprised by the results, I do indeed see some value in the method and I thereby recommend consider this manuscript for acceptance.

As a further suggestion the author may study some transfer learning properties, i.e., how well the correction NN would perform when being trained on a specific set of initial condition but used in different setups.

**Summary Of The Paper:**

The paper aims at deploying a deep learning approach in combination with partial differential equations (PDEs) with known yet incomplete physical information.
For example, a deep learning model can be employed when the information on the system to be solved is limited but additional data are available. A common benchmark is represented by turbulence modelling in computational fluid dynamics where the full PDEs descriptions are too expensive to be solved computationally. Therefore, the PDEs can be rewritten as a sum of an incomplete PDE description plus another term representing the physics arising from the unknown information (reaction rate terms). Correctly representing this additional term is of crucial importance to solve the task.

Contrarily to standard Neural PDE solvers, this method aims at combining physical incomplete PDE information with some additional machine learned correction due to the unknown part of the PDE description. In this work, the latter is represented by the chemical kinetic mechanism of non reactive flow simulation. The complete (reactive) simulation is used as a baseline.
The downstream task is to solve a simplified version of the Navier-Stokes equation for a specific system where the chemistry is removed from the analytic description (i.e., is unknown).

A neural network is trained to learn correction fields that would correct the incomplete (non reactive) PDE equations where both source terms and reaction rates have been removed.
The work compares three baselines: Ground truth, Pure Data Driven approach, Fourier Neural Operators (state-of-the-art neural operator method).

The framework is clearly explained in figure 2 where the PDD and the NN-Hybrid approaches are compared in B1 and B2 respectively.
The 3 different setups on which the approach was tested are carefully explained.

A few comments/concerns are in place:

- The following sentence is a bit unclear to me ‘Its objective is to learn to model the effects of the unknown chemistry using the neural network parameters θ given an input flow
 State […]’. What is it meant by ‘[…] using the neural network parameters \theta […]’?
- In the last row of table 1 some baselines are missing (i.e., FNO and NN.) I presume this is because they never converge to the target value? It would be helpful to write something about it in the table caption.
- On page 7 below figure 4, in the second paragraph the authors define the relative displacement. Right afterward, they mention x_t and \tilde{x}_0. I cannot relate these quantities easily to the definition and this may be overall a bit confusing. I would suggest to keep the definition of relative displacement general \tilde{x}_300 -> \tilde{x}_t and they specify that t=300. As it is now the discussion might not sound very linear and straightforward.
- I would suggest the authors to add additional plot to figure 6 to visualise the performance of other baselines like FNO and NN. From the discussion of table 1 is already clear they perform worse than NN-PDE, but visualising them (as it’s done in figure 12) might be useful.
- In section 5.3 the author only discuss the full NN method without mentioning the NN-PDE. I am wondering it that is intentional. If so, there’s a reason for that? Are there any limitations that prevent NN-PDE to be applied for this task?
- I suggest the authors to make the labels explicitly in figure 2 saying what’s the PDD and what’s the NN-hibridy approach for an easy first-sight intuition.
- The caption of figure 2 does not explain what is S. Though this is done in the main text would be useful to repeat it in the caption below the figure.
- I’d rename section 2 something like ‘Literature Review’ or ‘Related Work’. To me ‘Background’ sounds more like if you are introducing the physics behind the work.
- Missing full stop at the end of equation 3.
- Few typos/stylistic issues were found throughout the manuscript. I’d recommend the authors to run a spellchecker for making sure to detect them.

**Summary Of The Review:**

To incorporate physical prior information when training a neural network to solve a physical problem is known to be a successful approach.
In this paper the author propose to learn an incomplete description of a PDE and correct a posteriori using a NN for removing two terms which in general act on different time scales. Doing so, the PDE solver happens to be much more efficient in learning the dynamics of the incomplete system. Contrarily,  when learning the full dynamics of the complete PDE description (pure NN approach) the interplay between different time-scales makes the learning task much more difficult.

The idea of separating the PDE and learn part of the dynamics in a supervised learning fashion and correct the PDE result a posteriori is indeed a valuable and interesting contribution to the community.

Despite not being a domain expert, I still do see some potential in this work, especially in future developments.
For these reasons and all the considerations from above, I would recommend to consider this manuscript for acceptance.

---

> ### Author Response · Authors · 2022-11-18
> **Response to Reviewer hgQY**
>
> We thank the reviewer for the positive assessment of our work. We will discuss the updates made to address the questions and concerns in the following.
>
> **What is it meant by ‘[…] using the neural network parameters \theta […]’?**
>
> We denote the learnable parameters of the neural network model by $\theta$, which are adjusted at training time. The sentence has been rephrased to “The neural network is trained to model the effects of the unknown chemistry using parameters $\theta$ given an input flow state $\phi$.”
>
> **In the last row of table 1 some baselines are missing (i.e., FNO and NN.)**
>
> We study the nonUniform-Bunsen100 scenario with 100 × 100 resolution as a real-world scenario of Bunsen-type flame. As demonstrated from three different cases of 32 × 32 resolution with increasing difficulty, baselines FNO and PDD fail to capture the flame dynamics accurately. Therefore, we focus only on the hybrid NN-PDE approach for this highly resolved flame case. We have added these clarifications in section 5.1 of the paper.
>
> **Additional plot to figure 6 to visualise the performance of other baselines like FNO and NN**
>
> This is a good idea. Figure 13 compares predictions made by FNO, PDD and hybrid NN-PDE model for nonUniform-Bunsen32 case at twice the time-step ($2 \Delta t_c$), with the ground truth solutions coming from the complete PDE solver at time-step of $\Delta t_c$. FNO and PDD (upper two rows) fail to recover the correct flame shapes over 250 simulation steps. The hybrid NN-PDE model (second row from below) predicts the flame shape accurately, thus relaxing the temporal stiffness of the complete PDE solver.  Additionally figure 12 compares the performance of the baseline approaches using twice the time-step size of the complete solver, with hybrid NN-PDE model. For the Planar-v0 and uniform-Bunsen case, PDD catches up with the hybrid approach whereas the PDD approach completely fails to predict the dynamics of the nonUniform-Bunsen32 case. We further investigate the nonUniform-Bunsen32 case with larger time-steps (2 times, 4 times and 8 times) as shown in figure 5 (B). It shows MAPE of the predictions increases for larger time-steps. PDD achieves MAPE of 22.44 ± 16.30 % for $8 \Delta t_c$ setup as compared to the MAPE of 12.48 ± 10.41 % for $\Delta t_c$ setup. The PDD approach does not predict the dynamics accurately for any of the larger time-steps considered.
>
> **In section 5.3 , Are there any limitations that prevent NN-PDE to be applied for this task? As a further suggestion the author may study some transfer learning properties**
>
> Section 5.3 discusses potential application of the learned hybrid NN-PDE of reactive flows to control flame shapes. Using transfer-learning techniques, the learned hybrid NN-PDE can be successfully integrated to control the flame shapes. We’d be happy to add an additional demonstration of this combination in a future version of our paper. We have added these comments in the ‘Conclusions’ section.
>
> Additional minor corrections:
> - We have updated the definition of the relative displacement as suggested.
> - We have updated figure 2 to specify the prediction approach and have updated the caption to clarify notation ‘S’. ‘S’ represents the concatenation of various fields to obtain the updated flow field $\tilde{\phi}$ at next time step.
> - Section 2 is renamed as ‘Related Work’.

---

> > ### Comment · Reviewer_hgQY · 2022-11-23
> > **Acknowledgement to Authors' response**
> >
> > I thank the authors for addressing my concerns and extending the manuscript including the clarifications I asked for.
> > I also thank the author for the minor adjustments they made.
> >
> > I already found the paper quite nice and I don't have much to add to my previous rating.

---

### Official Review · Reviewer_c79Z · 2022-11-04

**Confidence:** 3
**Correctness:** 2
**Technical Novelty And Significance:** 1
**Empirical Novelty And Significance:** 3
**Recommendation:** 3

**Clarity, Quality, Novelty And Reproducibility:**

# Clarity
The paper's clarity is ok but should be improved to be accepted for publication, in my opinion. For instance, see my confusion regarding some of the points mentioned in the weaknesses of the paper.

# Quality
I did not find the experimental validation solid enough to trust the empirical conclusion made by the authors entirely. For example, the authors do not explain how they selected the different architectures and training hyperparameters.

# Novelty
The novelty is limited. The main contribution is to showcase a simple hybrid approach to original test scenarios.

# Typos and minor remarks
- What is u in eq (1)?
- The second paragraph before section 3: don't -> do not
- The paragraph before section 3: Why don't you discuss Takeishi and Kalousis 2021 (https://scholar.google.com/citations?view_op=view_citation&hl=fr&user=rqF9bAsAAAAJ&citation_for_view=rqF9bAsAAAAJ:hC7cP41nSMkC)? I am not an author of this paper, it is a genuine remark. I also wonder why you do not compare to an approach similar to APHYNITY (https://scholar.google.com/citations?view_op=view_citation&hl=fr&user=rFaxB20AAAAJ&sortby=pubdate&citation_for_view=rFaxB20AAAAJ:IZKZNMMMWs0C).
- Eq (3): a point at the end of the equation is missing.
- Eq (5): the upper limit should be t + m.
- unfiform-Busen -> Uniform-Busen.

**Strength And Weaknesses:**

# Strength
The idea of using the physical understanding of the studied phenomenon, rather than an entirely data-driven one, makes a lot of sense to me. In addition, the systems studied in this paper seem novel to the hybrid learning literature and constitute solid test scenarios for assessing future developments in hybrid learning.

# Weaknesses
Overall I find the contribution very limited. Combining a neural network with equations from physics is a long-standing idea in the community. This paper does not introduce anything subtle regarding a good strategy to combine these equations with neural networks.

The experimental validation of their approach also seems limited in terms of the train/test scenarios considered. In addition, it is unclear how the authors selected the best models and hyperparameters, as no cross-validation or validation set is discussed. This also reduces the solidity of the presented results. For instance, I am pretty sure that at some point, the data-driven approach could catch up with the hybrid approach if the number of train scenarios increases. An analysis of when this (might) happen would have been valuable to motivate the hybrid approach for these problems.

It is unclear to me whether the parameters of the PDE are also learned at training time. If not, this makes the proposed approach's applicability very limited as, in most cases, if we can observe a system and have little understanding of the physics behind this also means that we do not know the corresponding parameters of the incomplete physical description. I have probably misunderstood something there, but I would encourage the authors to clarify this in the paper. Overall, this paper would benefit from a real-world demonstration of the proposed approach. Indeed, the contribution is not methodological. Thus I would argue that this should contain a more solid empirical evaluation.

As a remark, I also feel that some confusion is made between the "model" and the "inference" (in the case of PDEs, the solver). The neural network aims to complete the model, create a more accurate description of the real world, and create a better model. The solver is related to how we make inferences given a model; the terms complete and incomplete solvers do not make sense to me.

**Summary Of The Paper:**

The paper argues for the combination of a neural network with an incomplete description of a system (a hybrid model), in their case, an incomplete PDE, rather than a fully data-driven approach that only relies on a neural network. They compare the two approaches on a set of systems described by thermodynamics and fluid dynamics.

The authors consider a supervised setting where the complete PDE, or solutions for it, are given and used for training the neural networks with an MSE loss. In this setting, their experiments suggest that the hybrid models outperform the data-driven approach by a significant margin. In addition, they argue that hybrid models can lead to faster solving of the PDEs than relying on the complete description when it is available.

**Summary Of The Review:**

Overall I recommend rejection of this paper for the aforementioned reasons. In particular, the lack of novelty of the approach and the limited reliability of the experiments.

---

> ### Author Response · Authors · 2022-11-18
> **Response to Reviewer c79z (1/2)**
>
> **At some point, the data-driven approach could catch up with the hybrid approach if the number of train scenarios increases.**
>
> Thank you for raising this interesting point. We have added a new experiment to study the effect of training dataset size on the accuracy of PDD predictions. We train PDD models with different numbers of simulations in the training dataset. Each simulation contains 500 time-steps. Figure 5 shows the MAPE of PDD models trained with {4, 8, 12, 16, 24, 32} training datasets, over a fixed testset. We compare the performance of these PDD models with an equivalent (trained with same look-ahead steps *m=2*) hybrid NN-PDE model, trained with 12 simulation sets. Increasing the number of training sets has little or no effect on the prediction capabilities of the PDD models. The hybrid model with 12 training sets achieves a MAPE of 12.49 ± 4.17%, an improvement of 38% over the PDD model with 32 training sets. This result strengthens the hypothesis that integrating the incomplete PDE solver into the neural network training yields a learning signal that fundamentally differs from that produced by training with precomputed data. The purely data-driven models cannot achieve the same level of accuracy even in the presence of large amounts of data.
>
> **It is unclear to me whether the parameters of the PDE are also learned at training time. In most cases, we do not know the corresponding parameters of the incomplete physical description.**
>
> The proposed hybrid NN-PDE model is capable of completing the PDE description even if the underlying incomplete PDE solver has incorrect parameters. We refer to the incomplete solver with incorrect parameters as ‘incomplete, incorrect PDE’. We ran an additional experiment wherein we combined an incomplete, incorrect PDE solver with a neural network model. We assume that the known values of the incomplete PDE parameters in equation 2, strain rate tensor ($\tau$), the diffusion coefficient of species k ($D_k$) and mixture thermal conductivity (𝜆), are incorrect. Figure 9 (A) shows the difference between temperature field evolution of the incomplete PDE solver with correct parameters ([$\tau$, $D_k$ ,𝜆] = [0.1, 0.1, 0.1]) and incorrect ([0.05, 0.05, 0.05]) parameters at different time-steps. The hybrid NN-PDE combines this incomplete, incorrect PDE solver with the neural network model to obtain the solutions of the complete PDE solver with correct parameters. Figure 9 (B) compares the flame dynamics predicted by this hybrid NN-PDE model with that of the complete, correct PDE solver. A good match is observed over various test cases. The hybrid model with incomplete, incorrect solver achieves an overall MAPE of 2.48 ± 1.20 %, compared to the MAPE of 2.04 ± 1.39 % for hybrid model with incomplete, correct PDE.
>
> In addition, including system parameters like the conductivity as variables in the learning problem would be an interesting topic for future work. We have not yet performed such tests, but the inverse problems of section 5.3 point to the fact that the proposed setup could deal with such cases.
>
> **Overall, this paper would benefit from a real-world demonstration of the proposed approach.**
>
> We would like to point out that various reactive flow cases and the treatment of the operating conditions demonstrated are closely related to real-world applications in the combustion community: for example the nonUniform-Bunsen32 and nonUniform-Bunsen100 cases with highly resolved flames are closely related to the widely studied (numerically and experimentally) Kornilov’s flame [Kornilov et al., CnF, 156(10), 1957-1970 (2009), Duchaine et al., CnF, 158(12), 2384-2394 (2011)]. Thus, we have chosen the simulation setup described in our paper because of its significant practical relevance.
>
> **As a remark, I also feel that some confusion is made between the "model" and the "inference" (in the case of PDEs, the solver). the terms complete and incomplete solvers do not make sense to me.**
>
> The terms incomplete and complete are used in relation to the system of equations (PDEs) and their associated solver is referred to as incomplete/complete PDE solver. We have changed the phrase ‘complete/incomplete solver’ to ‘complete/incomplete PDE solver’ in the manuscript.

---

> > ### Author Response · Authors · 2022-11-18
> > **Response to Reviewer c79z (2/2)**
> >
> > **The paragraph before section 3: Why don't you discuss Takeishi and Kalousis 2021 paper.**
> >
> > We apologize for not including these works in our previous manuscript. We have updated the section ‘Related work’ to discuss these papers. Yin et al. 2021 (APHYNITY) and Takeishi and Kalousis 2021 (Physics integrated VAEs) introduce frameworks of augmenting incomplete physical dynamics with neural network models to forecast the solutions of a complete dynamical system. These approaches demonstrate the applicability on ODE/PDE systems, which are weakly non-linear and the unknown dynamics are linear combinations of the underlying fields. We explore a significantly more challenging application: a reactive flow. Note that a reactive flow is characterized by a multi-physics system with non-linear advective terms and strongly nonlinear dynamics, described by exponential source terms that exhibit non-linear combinations of the flow fields. As such, we believe our work represents an important continuation of the line of work started by these two papers.
> >
> > From the APHINITY paper, it is not directly clear if it can be extended to our application of reactive flows. Our application requires a sophisticated differentiable fluid flow solver, such as PhiFlow to implement the incomplete PDEs.
> >
> > **The experimental validation of their approach also seems limited in terms of the train/test scenarios considered.**
> >
> > We have carefully designed train/test cases to test the prediction capabilities of the model. For Planar-v0 case, test cases considered exhibit different flame temperatures and displacements. For uniform-Bunsen case, flame temperature and flame length of test cases are different from the training dataset. For nonUniform-Bunsen case, flame shapes are considerably different from the training dataset. Appendix A and table 3, justifies the choice of neural network architecture used (ResNet vs. UNet). We have not performed hyper-parameter optimization for all the design choices made, but we have done an informal tuning of depth of the UNet architecture and CNN stack depth.

---

### Author Response · Authors · 2022-11-18
**General Response**

We want to thank all the reviewers for their efforts to review our submission and for the suggestions to improve our paper. All the reviewers found the complexity of the PDE systems studied novel and of practical relevance. To account for the questions and queries, we have uploaded a revised PDF version with changes highlighted in blue. We have extended our revision with several experiments and additions based on the requests. We consider them all valuable contributions and are very grateful for the suggestions.

The revised version includes the following major changes
- We have updated the ‘Related Work’ section to discuss how our work builds upon the direction established by Yin et al. 2021, and Takeishi and Kalousis 2021.
- We extended the evaluation of the PDD approach to study the effect of increased training dataset size.
- We have included additional results to study the effect of temporal coarsening on the performance of the baselines considered.
- We study the generalization of the proposed approach to incorrect PDE parameters in incomplete PDE solver.

---

### Decision · Program_Chairs · 2023-01-20

**Decision:**

Reject

**Justification For Why Not Higher Score:**

 Incremental novelty from the ML side and insufficient baseline comparisons

**Justification For Why Not Lower Score:**

N/A

**Metareview: Summary, Strengths And Weaknesses:**

The paper considers learning dynamical systems using hybrid ML + PDE solvers. They consider a setting were the governing equations are partially known and modeled by the solver, while the ML component complements for the unknown part. The hybrid model is motivated by a specific problem, the modeling of reactive flow systems, more precisely flame simulation. Experiments compare the proposed approach with pure ML approaches and show large improvement by incorporating the physics.

The question addressed in the paper has already motivated several contributions in the physics aware ML literature. The originality of the paper is to consider a problem more complex than the ones classically used in this literature. The approach demonstrates the relevance of the approach when compared to pure ML methods and even some benefits compared to pure simulation. The appreciation by the reviewers is mixed. This may be due to the chosen approach which does not develop a ML view point with for example the development of new ML-physics ideas, but focuses on solving a specific problem. The main concerns raised by some reviewers are the absence of comparison with closely related approaches, the incremental novelty from the ML side and insufficient baseline comparisons. Overall, one considers that the paper addresses an important question and has several merits but in its present form it is probably more adapted to an applied science venue than to a ML conference. The applicative scope should probably by widened for a submission to an ML audience.